# Orchestrated ensemble activities constitute a hippocampal memory engram

Khaled Ghandour [1,2,3,18], Noriaki Ohkawa[1,2,3,18], Chi Chung Alan Fung [2,4,16,18], Hirotaka Asai[1,2], Yoshito Saitoh[1,2,3], Takashi Takekawa [2,5], Reiko Okubo-Suzuki[1,2], Shingo Soya[6], Hirofumi Nishizono[2,7], Mina Matsuo[7], Makoto Osanai [8,9], Masaaki Sato[4,10,11], Masamichi Ohkura[10,11,17], Junichi Nakai[10,11], Yasunori Hayashi [4,11,12], Takeshi Sakurai[6], Takashi Kitamura [13,14], Tomoki Fukai [2,4,15] & Kaoru Inokuchi [1,2]

The brain stores and recalls memories through a set of neurons, termed engram cells. However, it is unclear how these cells are organized to constitute a corresponding memory trace. We established a unique imaging system that combines $Ca^{2+}$ imaging and engram identification to extract the characteristics of engram activity by visualizing and discriminating between engram and non-engram cells. Here, we show that engram cells detected in the hippocampus display higher repetitive activity than non-engram cells during novel context learning. The total activity pattern of the engram cells during learning is stable across post-learning memory processing. Within a single engram population, we detected several sub-ensembles composed of neurons collectively activated during learning. Some sub-ensembles preferentially reappear during post-learning sleep, and these replayed sub-ensembles are more likely to be reactivated during retrieval. These results indicate that sub-ensembles represent distinct pieces of information, which are then orchestrated to constitute an entire memory.

[1] Department of Biochemistry, Graduate School of Medicine and Pharmaceutical Sciences, University of Toyama, 2630 Sugitani, Toyama 930-0194, Japan. [2] CREST, Japan Science and Technology Agency (JST), Toyama 930-0194, Japan. [3] PRESTO, JST, 4-1-8 Honcho, Kawaguchi, Saitama 332-0012, Japan. [4] RIKEN Center for Brain Science, Hirosawa 2-1, Wako, Saitama 351-0198, Japan. [5] Faculty of Informatics, Kogakuin University, 1-24-2 Nishi-Shinjuku, Shinjuku-ku, Tokyo 163-8677, Japan. [6] WPI-IIIS, University of Tsukuba, 1-1-1 Tennodai, Tsukuba, Ibaraki 305-8575, Japan. [7] Life Sciences Research Center, University of Toyama, Toyama 930-0194, Japan. [8] Tohoku University Graduate School of Medicine, 2-1 Seiryo-machi, Aoba-ku, Sendai 980-8575, Japan. [9] Graduate School of Biomedical Engineering, Tohoku University, Sendai 980-8575, Japan. [10] Graduate School of Sciences and Engineering, Saitama University, 255 Shimo-Okubo, Sakura-ku, Saitama 338-8570, Japan. [11] Brain Body System Sciences Institute, Saitama University, Saitama City, Saitama 338-8570, Japan. [12] Graduate School of Medicine, Kyoto University, Yoshida-Konoe-cho, Sakyo-ku, Kyoto 606-8501, Japan. [13] Department of Psychiatry, University of Texas Southwestern Medical Center, 5323 Harry Hines Boulevard, Dallas, TX 75390-9070, USA. [14] Department of Neuroscience, University of Texas Southwestern Medical Center, 5323 Harry Hines Boulevard, Dallas, TX 75390-9070, USA. [15] Neural Coding and Brain Computing Unit, Okinawa Institute of Science and Technology, 1919-1 Tancha, Onna-son, Kunigami-gun, Okinawa 904-0495, Japan. [16] Present address: Neural Coding and Brain Computing Unit, Okinawa Institute of Science and Technology, 1919-1 Tancha, Onna-son, Kunigami-gun, Okinawa 904-0495, Japan. [17] Present address: Graduate School of Clinical Pharmacy, Kyushu University of Health and Welfare, Nobeoka 882-8508, Japan. [18] These authors contributed equally: Khaled Ghandour, Noriaki Ohkawa, Chi Chung Alan Fung. Correspondence and requests for materials should be addressed to N.O. (email: nohkawa@med.u-toyama.ac.jp) or to K.I. (email: inokuchi@med.u-toyama.ac.jp)

In the brain, memories are thought to be stored within sub-populations of neurons, termed engram cells, which are activated during learning[1,2]. Activity in these cells corresponds to a specific event and ensures recovery of that experience[3–7]. Optogenetic activation[3,5,8] or inhibition[9] of engram cells induces or inhibits memory recall, respectively. Thus, engram cells represent the physiological manifestation of a specific memory trace. These subsets of neurons mirror experiences in our life and can be identified through expression of the immediate early gene, *c-fos*[3,10]. Neuronal memory ensembles are formed by the synchronous activity of cells within a certain time window in response to a cognitive event[11,12]. However, technical limitations mean that it is difficult to distinguish the activity of engram cells from that of non-engram cells in real time. Therefore, it is not clear how activity in these engram cells is assembled to represent the corresponding event.

There is growing consensus about the importance of replay activity, which is the offline reactivation of the previous activity. Replay activity is thought to be critical for conserving and consolidating encoded events[13]. During post-training rest and sleep, spatial memories are disrupted by selective inhibition of the sharp wave ripples within which sequential activity from a prior experience reappears[14]. Similarly, reduced theta power during rapid eye movement (REM) sleep weakens previously acquired hippocampus-dependent memories[15]. These findings imply that, during the post-learning period, specific activity in a certain group of cells represents and governs the maintenance of a given memory trace. However, observing and tracking such cells in real time across the memory processing stages is necessary to clarify the properties of that activity.

In the hippocampus, CA1 neurons have the ability to fire in a synchronous and coordinated manner, forming cell ensembles[12,16]. The activity within a coherent group of neurons, ensembles, rather than in single neurons, may provide a better representation of a certain event[11]. The orchestration of neural ensembles is likely the counterpart for complex information. Recent techniques such as $Ca^{2+}$ imaging provide an opportunity to observe and track the activity of a large number of neurons simultaneously over long time periods in freely behaving mice[17,18], enabling the functional importance of these assemblies during the post-learning period to be decoded.

In this study, we used an imaging system with a miniature head-mounted fluorescent microscope[17,19,20] to identify engram cells using the photoconvertible fluorescent protein Kikume Green Red (KikGR)[21] and the c-fos-tet-tag system[22]. We monitored neuronal activity in the CA1 hippocampal area via $Ca^{2+}$ influx measured with G-CaMP7[23]. We show that contextual memory in the hippocampus is represented as distinct subsets of synchronous activity (defined by $Ca^{2+}$ transients) that comprise several ensembles of engram cells. In contrast to non-engram cells, these ensembles maintain their activity not only during learning but also during post-learning sleep and retrieval sessions.

## Results

### Labelled cells in CA1 induces recall of contextual memory. 

To understand how contextual memories are represented in engram cells, we first confirmed that cells labelled with c-fos during contextual exposure do indeed store the contextual memory. Engram cells can be specifically targeted in c-fos-tTA mice because the neural activity associated with memory formation induces c-fos expression, which in turn induces activity-dependent tTA expression under the control of the c-fos promoter. In the absence of doxycycline (Dox), tTA binds to the tetracycline-responsive element (TRE), enabling downstream expression of the TRE-dependent transgene[3,22,24]. We injected lentivirus (LV)-expressing channelrhodopsin-2 (ChR2) or enhanced yellow fluorescent protein (EYFP) under the control of TRE in c-fos-tTA transgenic mice[5]. This system ensures that only the neurons in the ensemble specific to a certain event will be labelled with ChR2 or EYFP. We took advantage of a context pre-exposure facilitation effect (CPFE) paradigm[25,26] to investigate contextual memory per se. Mice explored a novel context under Dox removal (OFF Dox), during which the context engram was labelled. One day later, they received a foot shock in the same context (paired) or a different context (unpaired) during ON Dox, which prevented any additional labelling of cells associated with the shock information (Fig. 1a). In this paradigm, mice can associate context A (the pre-exposure context) with a temporally separated immediate shock (IS) delivered in the same context. Therefore, re-exposure of the mice to the original context induces freezing behaviour in the paired, but not the unpaired, group (Fig. 1b). This result clearly demonstrates that pre-exposed-context information is associated with fear information only in the paired group. Optogenetic activation at 4 Hz of the CA1-labelled neurons[8] in a neutral context retrieved the fear memory in the ChR2 paired group, but not in the ChR2 unpaired or the EYFP group (Fig. 1c), even though all groups showed comparable levels of labelling efficacy (Fig. 1d, e). These results indicate that the labelled engram cells represent and encode the specific contextual memory.

### In vivo $Ca^{2+}$ imaging of engram cells identified with KikGR. 

To investigate how memories are represented and consolidated, we need to visualize the activity of engram and non-engram cells in real time. Thy1-G-CaMP7 mice express a $Ca^{2+}$ indicator, G-CaMP7, preferentially in pyramidal neurons in the deep layer of hippocampal CA1[23]. We developed a technique that combines a head-mounted, miniature fluorescent microscope[17,19] with Thy1-G-CaMP7/c-fos-tTA double transgenic mice. The hippocampal CA1 region in double transgenic mice was injected with LV expressing a fluorescent protein, KikGR[21], under the control of TRE (Fig. 2a). KikGR undergoes rapid and highly efficient photoconversion from green to red upon exposure to 365 nm light, rendering it invisible during subsequent $Ca^{2+}$ imaging (Fig. 2b and Supplementary Fig. 1A). Using this approach, we labelled the activated engram with KikGR and tracked the $Ca^{2+}$ signals corresponding to the activity of engram cells during contextual learning and post-learning sleep (Fig. 2b). Cells labelled with KikGR were visualized in a snapshot taken 24 h after context learning. Using regions of interest (ROIs) for KikGR-positive cells prepared from this snapshot, we were able to discriminate engram cells from non-engram cells (Fig. 2b and Supplementary Fig. 1B, C). To track the activity of both engram and non-engram cells through different memory stages, it was extremely important to make sure that we were tracking the same cells across different days. Therefore, we first processed each single $Ca^{2+}$ movie (binning, motion correction, removal of background) from each recorded session separately (Fig. 2c; see Methods). Then, all recorded sessions were concatenated, and we ran the motion correction one more time using a single frame to make sure that there was no change in XY translation across the entire movie, and hence no change in the spatial location of the recorded cells (Fig. 2c). To check the accuracy of the motion correction, we selected several representative cells from both the engram and non-engram groups that were highly active on both days and verified that there was no change in their spatial location (Fig. 2d, e). For some cells, the shape of the cell with dendritic shaft was apparent on both day 1 and day 2, further confirming that the spatial location of these cells was unchanged (Fig. 2e and Supplementary Fig. 1D, E). These results indicate that our system can

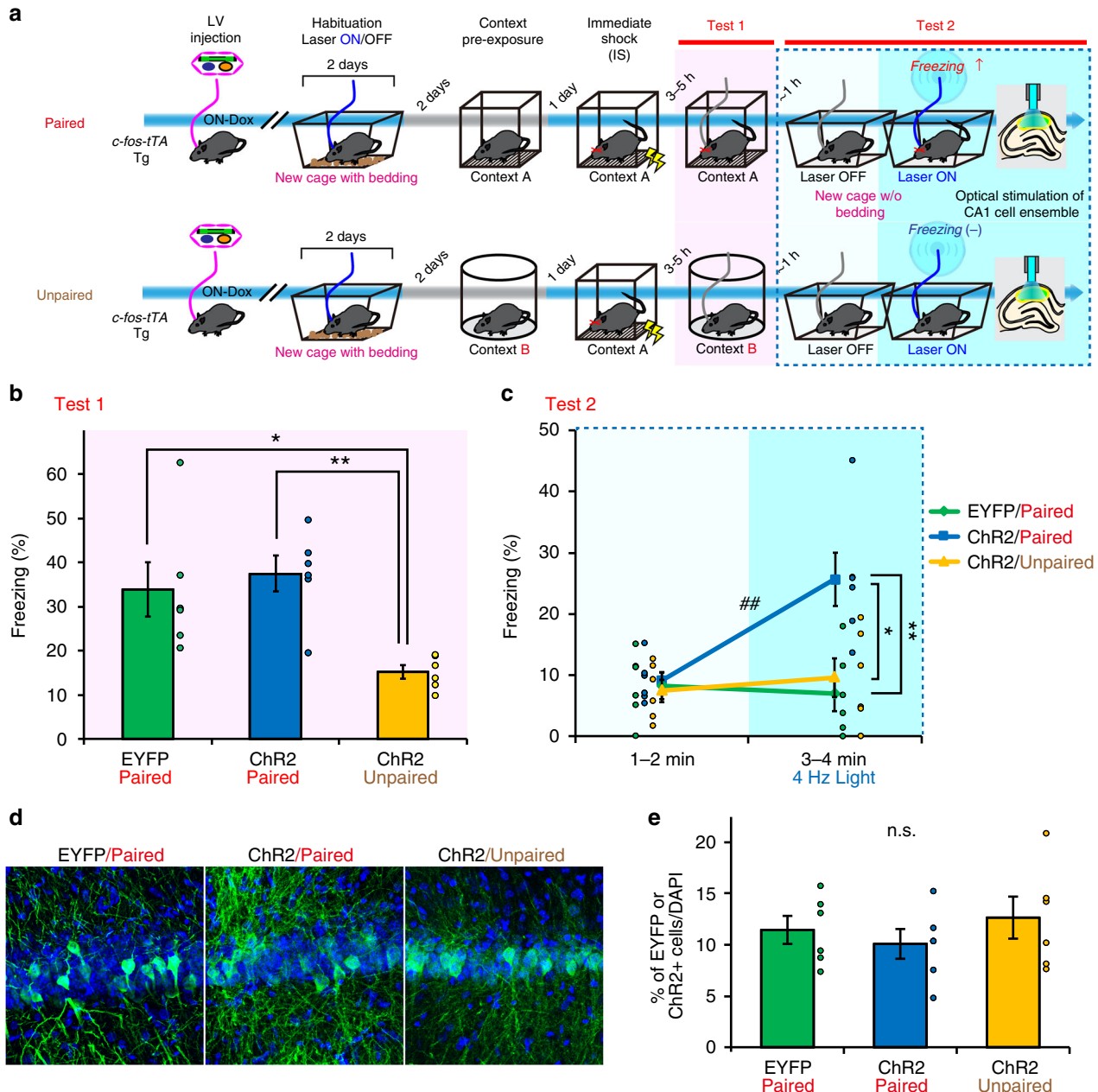

**Fig. 1** Optogenetic activation of c-fos-Tet-tagged cells in CA1 induces recall of a contextual memory. **a** Schematic diagram showing the behavioural paradigm for labelling the engram corresponding to contextual information, but not shock information, using the context pre-exposure facilitation effect (CPFE) paradigm (see Methods). Blue and grey lines represent ON and OFF doxycycline (Dox), respectively. **b** Test 1 shows significantly higher levels of freezing in the paired channelrhodopsin-2 (ChR2) and enhanced yellow fluorescent protein (EYFP) groups (one-way analysis of variance (ANOVA), $F_{(2,17)} = 7.46915$, $p = 0.00561$; Scheffe's post-hoc test, $*p < 0.05$, $**p < 0.01$) than in the unpaired ChR2 group. This indicates that information about context A was associated with information about the shock only in the paired group. **c** Test 2 showed comparable freezing in all groups during the first 1–2 min ($p > 0.82$, one-way ANOVA) when the light was off. There was a significant increase in freezing in the ChR2 paired group, but not in the EYFP paired or ChR2 unpaired groups, upon 4 Hz light stimulation during the 3–4 min after onset of test 2 (two-way repeated-measures ANOVA, $F_{(2,35)} = 8.64192$, $p = 0.003186$; Scheffe's post-hoc test, $*p < 0.05$, $**p < 0.01$; paired $t$ test, two-tailed (EYFP paired) $t_{(5)} = 0.38532$, $p = 0.71$, (ChR2 paired) $t_{(5)} = -4.90136$, $p = 0.0044$, (ChR2 unpaired) $t_{(5)} = -0.74587$, $p = 0.489$, $^{##}p < 0.01$). **d** Representative image of EYFP/ChR2 expression in the CA1 of lentivirus (LV)-injected c-fos-tTA mice. Green and blue signals represent EYFP and 4′,6-diamidino-2-phenylindole (DAPI) nuclear staining, respectively (scale bar, 50 μm). **e** The percentage of cells labelled with EYFP/ChR2 per DAPI staining was comparable across the test group and control groups (one-way ANOVA, $F_{(2,17)} = 0.63404$, $p = 0.544$). EYFP paired group, ChR2 paired group, and ChR2 unpaired group: $n = 6$ each. Data represent the mean ± s.e.m. n.s. not significant

separate engram and non-engram cells and track their activity in successive recording sessions.

**Engram cells show high repetitive activity in novel context.** Calcium transients were automatically identified by an

automatic sorting system, HOTARU[27], and then cells were categorized into engram and non-engram groups on the basis of the snapshot of KikGR expression taken 24 h after context exposure (Fig. 3a, Supplementary Fig. 1C and Supplementary Table 1; see Methods). Calcium transients obtained by

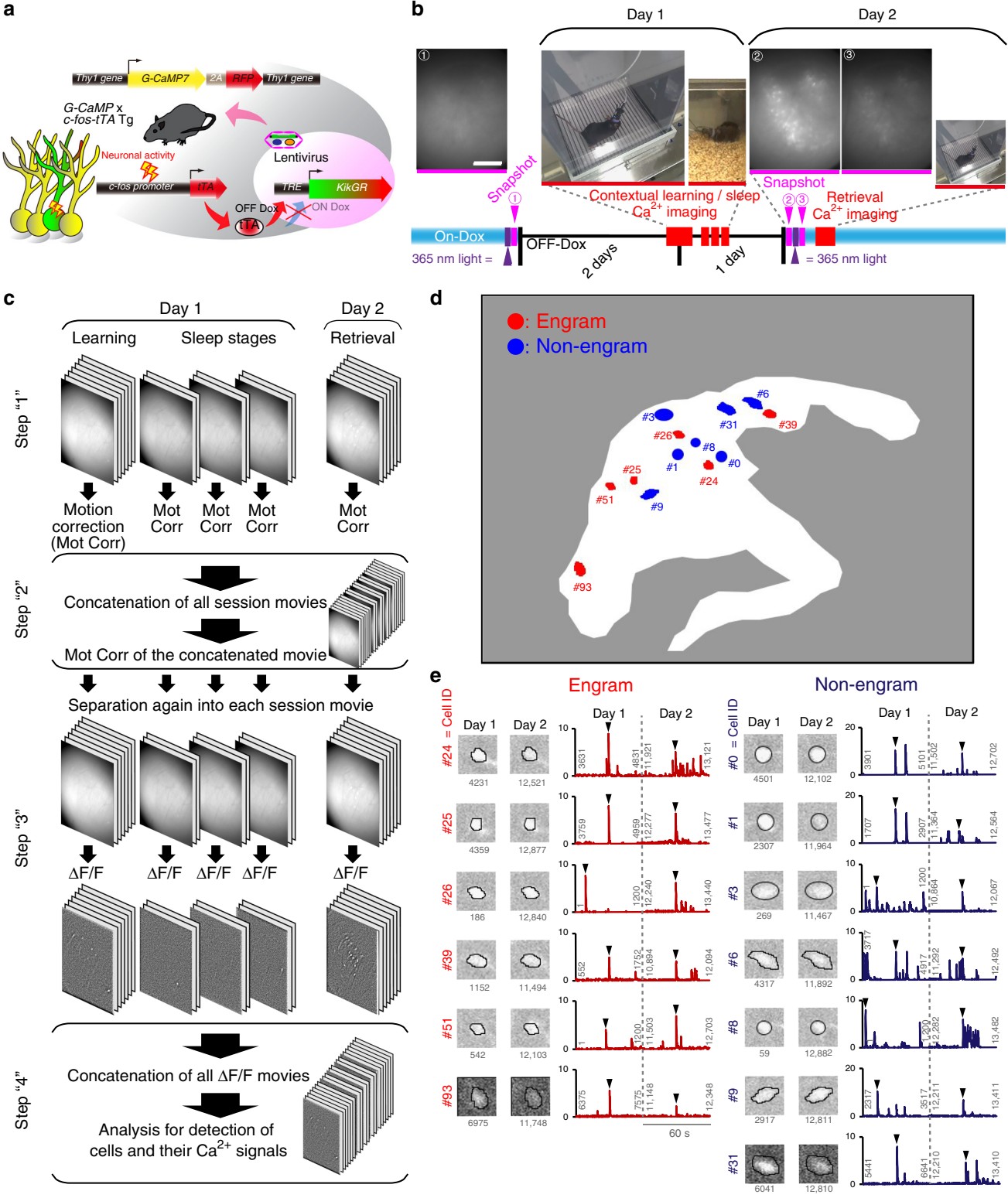

HOTARU were comparable to those obtained by constrained non-negative matrix factorization (CNMF-E) (Supplementary Fig. 2)[28]. We compared the activity patterns of engram and non-engram cells. Correlation matrix analysis showed that the activity observed in engram cells was repeated more frequently during novel contextual learning than either the activity in non-engram cells or the shuffled activity from engram cells (Fig. 3b,

c). Repetitive activity was quantified by temporal correlation matrix analysis[29]: the sum of all correlations during the first 60 s of contextual learning was significantly higher for engram cells than for non-engram and shuffled engram cells (Fig. 3d). These results indicate that engram cells exhibit the characteristic trait of highly repetitive activity during memory encoding.

**Fig. 2** In vivo Ca$^{2+}$ imaging of engram cells identified with the photoconvertible protein, Kikume Green Red (KikGR). **a** Schematic diagram showing labelling and visualization of engram and non-engram cells in Thy1-G-CaMP7 × c-fos-tTA double transgenic mice injected with TRE-KikGR lentivirus (LV). **b** Experimental design for calcium imaging. Top: snapshot of KikGR (scale bar, 200 μm), before (1) and after (2) contextual exploration, and after photoconversion (3), and representative photographs of behaviour during contextual learning, sleep and retrieval. Bottom: Ca$^{2+}$ imaging paradigm. Photoconversion by 365 nm light (dark purple arrow) and snapshot (pink arrow) before OFF doxycycline (Dox), then calcium imaging during contextual learning and sleep with OFF Dox, followed by a second snapshot and photoconversion after reinstatement of ON Dox, then a calcium imaging session during retrieval. **c** Schematic diagram showing the processing of the Ca$^{2+}$ imaging for the entire movie. Step 1: binning, motion correction, and background removal for each session independently; Step 2: concatenation of all movies into one long movie; motion correction run on the entire movie using a single reference frame; Step 3: separation of the entire movie into single sessions; $\Delta F/F$ performed on each movie; Step 4: re-concatenation of the single movies and procession to auto-detection of Ca$^{2+}$ signals. Mot Corr, motion correction. **d** Field of view (FOV) from a representative animal; white enclosed area is the area that showed KikGR expression and was incorporated in the analysis, with some representative engram and non-engram cells indicated in red and blue, respectively, together with their ID labels. **e** Regions of interest (ROI) of the labelled cells and their representative traces (extracted between the frames indicated) on days 1 and 2 for both engram cells (left) and non-engram cells (right). Arrowheads point to the time selected for evaluating the cell location. The number of frames is indicated under each ROI

**The total activity of engram is stable across sessions**. To measure the activity of engram cells through different memory processing stages from learning through post-learning sleep to retrieval, we recorded Ca$^{2+}$ transients during a contextual learning session (session A) as well as during post-learning sleep sessions comprising two non-rapid eye movement (NREM) sessions, NREM1 (session B) and NREM2 (session C), separated by at least several minutes, and one REM session proximal to learning (session D) (Fig. 4a). Sleep stages were discriminated on the basis of electroencephalogram (EEG) and electromyogram (EMG) traces (Fig. 4b). On the following day, mice were exposed to the same context as session A for memory retrieval (session E) and then, about 2 h later, to a different context (session F) (Fig. 4a).

We constructed population vectors for the activity of engram and non-engram cells in each session to assess the similarity of the population activity across sessions. Then, we measured the Mahalanobis population vector distance (PVD)[20], which measures the distance between two population vectors while taking the spread (covariance) of data points into account (see Methods). To investigate how neuronal activity changed across sessions, we measured the mean PVD of each session with respect to the learning session (session A). PVD is proportional to the number of dimensions (number of cells in this case). Given that the number of non-engram cells recorded was nearly 10-fold greater than the number of engram cells, it would not be fair to compare the PVD between engram and non-engram groups without controlling for the number of dimensions. Therefore, we computed a reduced Mahalanobis distance, where the reduced distance sums only the eigenvalues that contributed the most to the original Mahalanobis distance (see Methods). Notably, the engram cell activity was stable across sessions, as demonstrated by only small changes in the PVD across sessions (Fig. 4c, d). Thus, the population activity of engram cells in later sessions remained similar to the population activity observed during session A. By contrast, non-engram activity was more varied overall, with decreased response similarity to the learning session (A) (Fig. 4c, d), indicating that the population activity in non-engram cells was more diverse across sessions.

**Sub-ensembles construct a single engram population**. Given the repetitive activity during learning and the stability of the total activity across memory processing in engram cells, we next examined the temporal aspects and other components of this activity in more detail. Non-negative matrix factorization (NMF)[30] decomposes population activity into a time series of coactivated neuronal ensembles, termed "patterns" in this study (Fig. 5a). An optimal set of population activity patterns was selected on the basis of the Akaike information criterion with a second-order correction (AICc) to consider error function to approximate the data matrix by two non-negative factors, the pattern matrix and the intensity of the pattern (Supplementary Fig. 3A)[31]. Each pattern is composed of the different contributions from individual cells to make their synchronous activity; therefore, distinct patterns have different cell compositions and temporal activity, even among the group of engram cells associated with a single event (Fig. 5b). Neurons assigned to different ensembles were spatially intermingled (Fig. 5c). The intensity of each activity pattern calculated by NMF closely reflected the strength of the calcium transients of individual cells at a given time (Supplementary Fig. 3B, C). These results suggest that the total information in an engram is structured into sub-engram ensembles.

We conducted NMF analysis separately for engram and non-engram cells in each recording session to obtain separate sets of activity patterns for each session (sessions A to F; see Methods). Then, to determine the similarity of the patterns detected from different sessions, we calculated the cosine similarity (normalized dot product) between pattern vectors in different sessions. If two pattern vectors fully share the same cell composition and contribution, the normalized dot product score is 1. If there is no similarity, the score is 0. The results revealed that similar neuronal ensembles were repeated across different sessions in engram cells (Supplementary Figs. 4A, 5A), where a pattern pair was regarded as similar if the normalized dot product was >0.6. By contrast, very few ensembles detected in non-engram cells were repeated (Supplementary Figs. 4B, 5B).

Similar results were obtained by comparing the matching score (MS) between session A and the other sessions; MS(X,Y) measures the fraction of patterns in session X that are similar to any pattern in session Y. Thus, the MS gives a general representation of the similarity of the patterns detected in two sessions (see Methods). Remarkably, patterns detected in engram cells showed prominent, high MS values across sessions, with lower values in session F (where the animal was placed in a different context) (Fig. 5d, f). By contrast, MS values for non-engram cells were extremely low, with low similarity among patterns across sessions (Fig. 5e, f). Engram cells exhibited around 40–50% similarity between session A and sessions B, C, D and E. In marked contrast, this similarity dropped to around 10% in non-engram cells. This difference between engram cells and non-engram cells persisted across sessions (Fig. 5f, g). Comparable results were obtained when a higher threshold was implemented (Supplementary Fig. 5C). The MS dropped significantly from session E to session F only in the engram cells. Additionally, engram cells maintained their high MS across sessions when compared with engram shuffled data, except for sessions C and F (Fig. 5h, i). These results further confirm the stability and

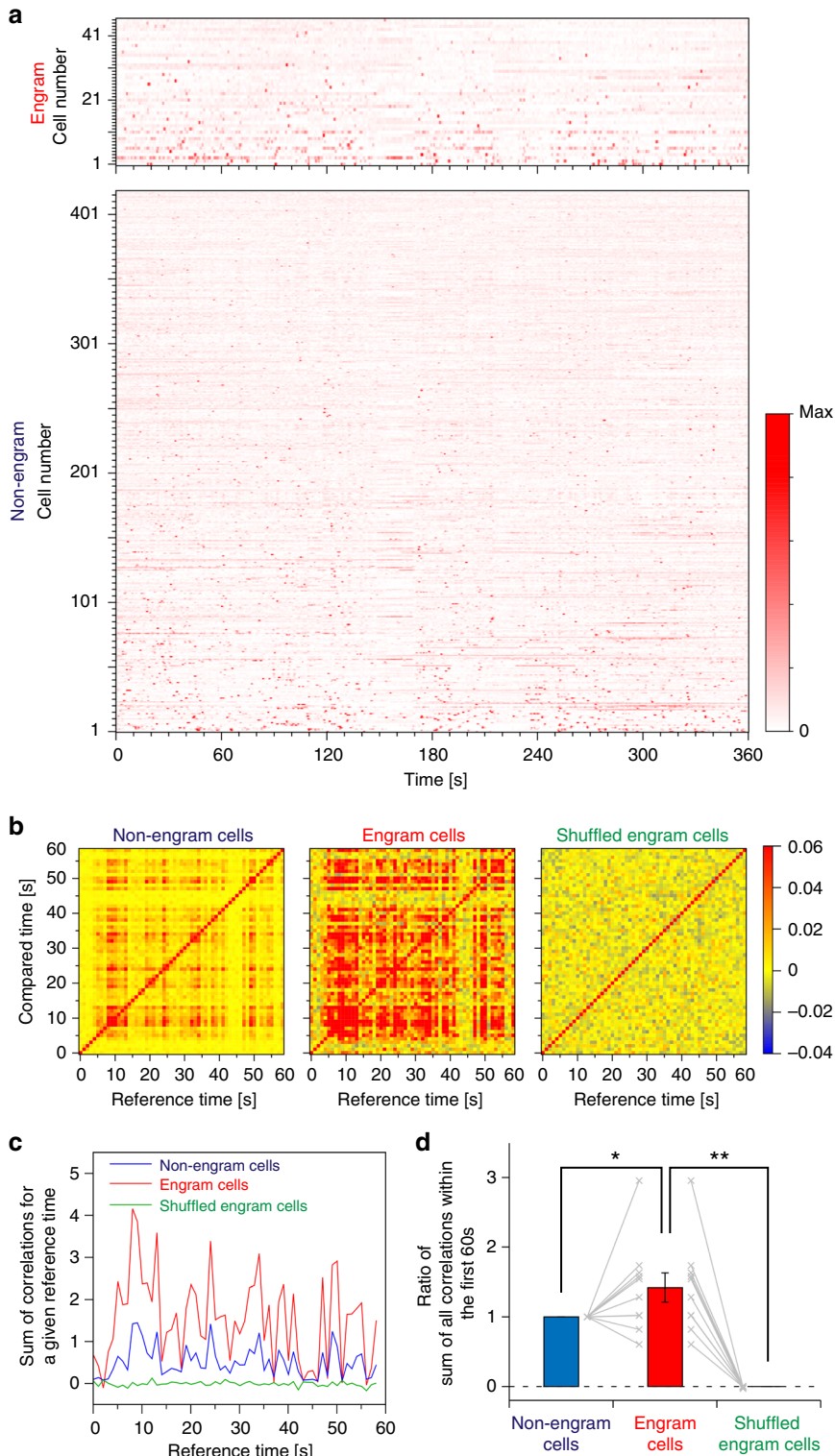

**Fig. 3** Engram cells show characteristic activity during contextual learning. **a** Normalized calcium transients for all engram cells (top) and non-engram cells (bottom) during contextual exposure (displayed over time). **b** Typical overlap $M(t, t')$ between correlation matrices at two time points (see Methods) for non-engram (left), engram (middle) and shuffled engram (right) cells within the first 60 s of calcium imaging. **c** Temporal sum of overlaps $M_{tot}(t)$ (see Methods) from a representative animal. **d** The average sum of overlaps in engram cells and shuffled engram cells, that is, the summation of $M_{tot}(t)$ over time during the first 60 s of calcium imaging relative to the sum from non-engram cells. Statistical comparisons were conducted using the Wilcoxon's signed-rank test, two-tailed. $n = 10$; (engram vs. non-engram) $t_{(9)} = 2.01$, $p = 0.037$ (engram vs. shuffled engram) $t_{(9)} = 6.81$, $p = 0.000038$; $*p < 0.05$, $**p < 0.01$. Data represent the mean ± s.e.m. across all animals

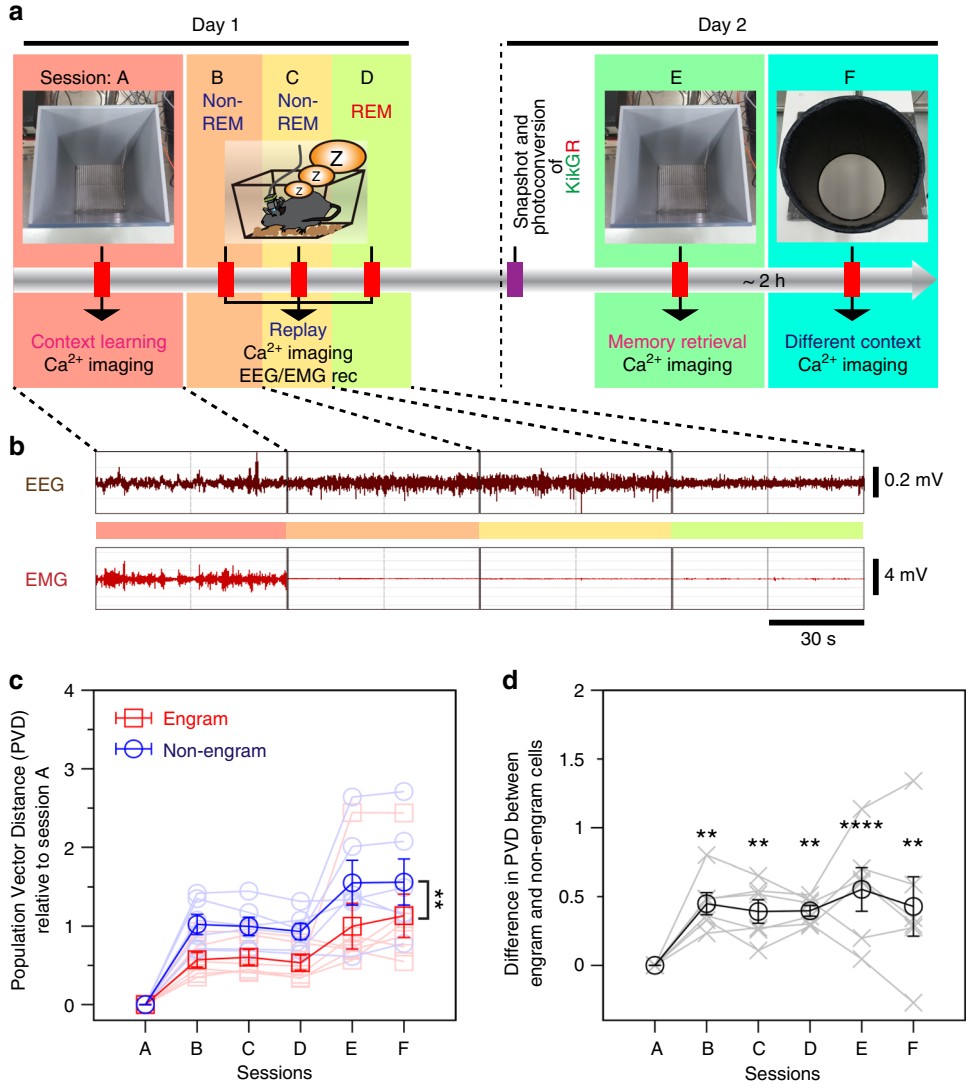

**Fig. 4** Total activity of engram cells is stable across memory processing. **a** Schematic diagram showing the calcium imaging paradigm.
**b** Electroencephalogram (EEG) and electromyogram (EMG) recordings while the animal was awake and during non-rapid eye movement 1 (NREM1), NREM2, and rapid eye movement (REM) sleep are indicated by red, orange, yellow and green, respectively. **c** Population vector distance (PVD) across sessions with respect to the learning session (A) in engram and non-engram cells. $n = 6$, two-way repeated-measures analysis of variance (ANOVA), $F_{(1, 5)} = 23.40$, $p = 0.0047$ (engram vs. non-engram). **d** Difference in PVD between engram and non-engram cells [$n = 6$; Bonferroni's multiple comparisons test, $p = 0.001$ (Session B), $p = 0.0043$ (Session C), $p = 0.0038$ (Session D), $p \leq 0.0001$(Session E), $p = 0.0017$ (Session F)]. $**p < 0.01$, $****p < 0.0001$. Data represent the mean ± s.e.m

repeatability of the activity of engram cells across sessions in the form of synchronously active subsets of neurons. Thus, ensembles formed during session A were more frequently reactivated in engram cells, but not non-engram cells, across post-training sleep and during retrieval, but not in a different context. Even after normalizing both engram and non-engram data by their shuffled activity, engram cells maintained their higher MS across sessions (Supplementary Fig. 6).

**Reactivated ensembles in sleep are reactivated in retrieval**. We next tracked each pattern observed in session A individually across the following sessions. This approach revealed that some of the patterns survived through sleep sessions B, C and D until session E (memory retrieval). These are termed aligned patterns (Supplementary Fig. 5). Some patterns were activated just once and were not detectable again in the next session, and these are termed isolated patterns (Supplementary Fig. 5). Pattern networking, in which nodes with pattern overlap larger than 0.6 are

connected, showed that engram cells had fewer isolated and more aligned patterns than non-engram cells. Aligned patterns constituted around 40% of the total number of patterns that appeared in session A in engram cells, while isolated patterns constituted around 80% of patterns in non-engram cells, regardless of whether NREM (Fig. 6a) or REM (Fig. 6b) sleep stages were considered. Thus, engram ensembles formed during the learning session and that were reactivated during sleep sessions (NREM or REM) were mostly reactivated during the retrieval session. By contrast, most non-engram ensembles that were activated during the learning session were not reactivated in the later sessions. Notably, half of the sub-ensembles from engram cells that were active in session A, sleep and retrieval were not active in session F (Supplementary Fig. 7), indicating that these sub-ensembles represent context A-specific memory.

Our model introduces the idea that an engram population corresponding to a memory event comprises several sub-ensembles that represent and replay the ongoing experience

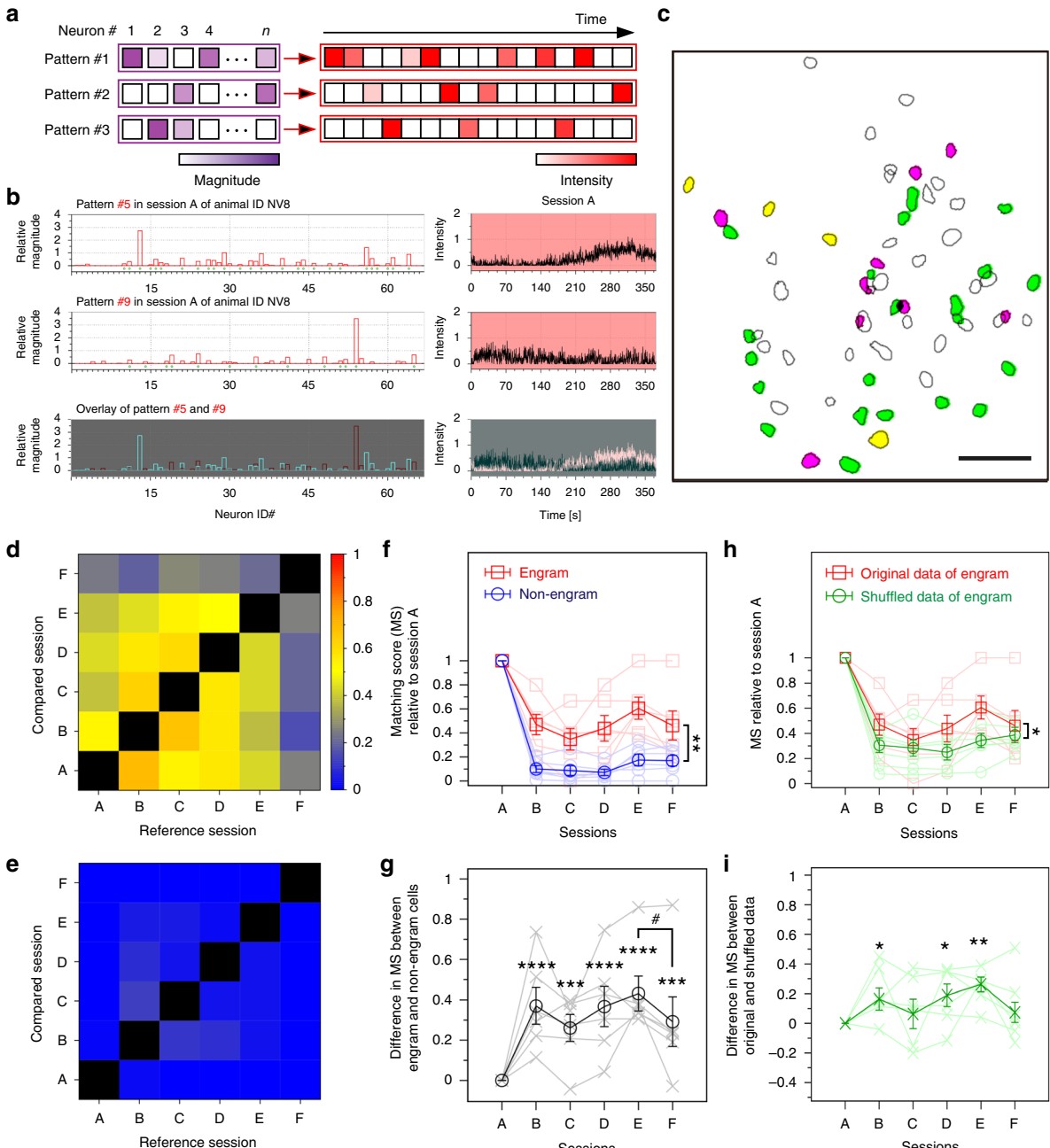

**Fig. 5** Contextual information is constructed by several sub-ensembles within a single engram population. **a** Diagrammatic representation of non-negative matrix factorization (NMF) analysis; coloured bars indicate the relative activity of each neuron in the detected pattern (shown as the magnitude) (left), and the relative intensity of the synchronous activity of this pattern across time (right). **b** Example of two detected patterns, with the participating neurons in each pattern (top and middle left) and their overlay (bottom left), and the temporal pattern of the corresponding intensity of synchronous activity during the learning session (A) (top and middle right) and their overlay (bottom right). **c** Distribution map of detected engram cells. Green- and magenta-filled contours indicate neurons assigned to two different ensembles (corresponding to patterns #5 and #9, respectively), and yellow-filled contours denote cells shared between patterns (scale bar, 100 μm). **d**, **e** Representative matching scores (MS) for pattern similarity analysis in engram cells (**d**) and non-engram cells (**e**). **f** MS across sessions (with respect to session A) for engram and non-engram cells. $n = 6$, two-way repeated-measures analysis of variance (ANOVA), $F_{1, 5} = 16.53$, $p = 0.0097$ (engram vs. non-engram). **g** Difference in MS between engram and non-engram cells across sessions [$n = 6$; Bonferroni's multiple comparisons test, $p \leq 0.0001$ (session B), $p = 0.0005$ (session C), $p \leq 0.0001$ (session D), $p \leq 0.0001$ (session E), $p = 0.0001$ (session F); paired $t$ test, one-tailed (E vs. F) $t_{(5)} = 2.47$, $p = 0.0284$]. **h** MS across sessions (with respect to session A) for original and shuffled engram data. $n = 6$, two-way repeated-measures ANOVA, $F_{(1, 5)} = 8.657$, $p = 0.0322$ (original vs. shuffled engram). **i** Difference in MS between original and shuffled engram data across sessions [$n = 6$, Bonferroni's multiple comparisons test, $p = 0.0122$ (session B), $p \geq 0.9999$ (session C); $p = 0.0296$ (session D), $p = 0.0017$ (session E), $p = 0.931$ (session F)]. $*p < 0.05$, $**p < 0.01$, $***p < 0.001$, $****p < 0.0001$, $^{\#}p < 0.05$. Data represent the mean ± s.e.m

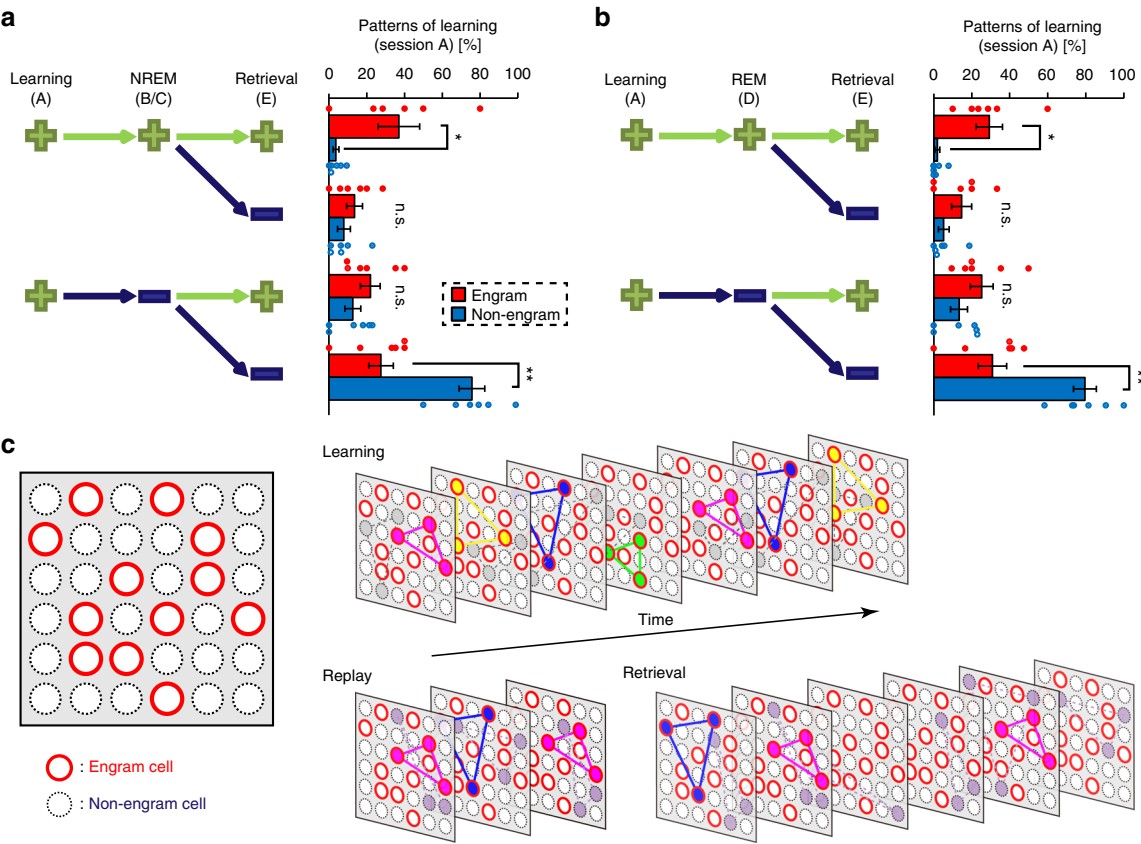

**Fig. 6** Engram sub-ensembles are frequently reactivated during memory processing. **a**, **b** Pair-making possibilities of each pattern expressed during learning with other patterns expressed from the non-rapid eye movement (NREM) (**a**) or rapid eye movement (REM) (**b**) sessions to retrieval. Positive green marks (+) and negative dark blue marks (−) indicate the presence or absence, respectively, of a significant pattern pair in the corresponding session. Pattern pairs with a dot product >0.6 are considered similar. Graphs show the percentage of patterns relative to the total number of patterns that appeared in the learning session. Statistical comparisons between engram and non-engram sub-ensembles were conducted with a paired $t$ test, two-tailed. $n = 6$; (+++) vs (+++), $t_{(5)} = 2.882$, $p = 0.03$; (+−−) vs. (+−−), $t_{(5)} = 4.53$, $p = 0.006$ in **a**; (+++) vs. (+++), $t_{(5)} = 3.763$, $p = 0.013$; (+−−) vs. (+−−), $t_{(5)} = 4.654$, $p = 0.0056$ in **b**; *$p < 0.05$, **$p < 0.01$, n.s. not signficant. Statistical comparisons between engram sub-ensembles reactivated during sleep and retrieval vs. during retrieval, but not sleep were conducted with a paired $t$ test, two-tailed. NREM, $p = 0.34$; REM, $p = 0.74$. Data represent the mean ± s.e.m. **c** Diagrammatic model showing orchestration of ensembles of engram cells across memory processing

during post-learning periods such as NREM and REM sleep. Replayed ensembles are more likely to be reactivated during a retrieval session (Fig. 6c).

## Discussion

Engram cells are activated upon exposure to a particular event, and it has been demonstrably shown that these subsets of neurons can retrieve the memory of that event upon reactivation. Monitoring the activity of engram cells in vivo has proved quite challenging because of the technical difficulties associated with observing and tracking the activity of both engram and non-engram cells across the entirety of memory processing from encoding, through consolidation, to retrieval. This study provides deeper insights into the neuronal activity of engram cells, at the population level, by taking advantage of a unique system that combines G-CaMP7-mediated $Ca^{2+}$ imaging and discrimination of engram and non-engram cells by a photoconvertible fluorescent protein, KikGR.

Using this system, we found that, compared with non-engram cells, engram cells exhibit a characteristic trait during novel context exposure: highly correlated repetitive activity. This suggests that engram cells tend to work in a synchronized, coordinated manner, through repetitive activity, to represent the ongoing experience. This is in line with the idea that engram cells fire in repetitive theta-modulated bursts during encoding[32]. It is also consistent with previous findings that engram cells have a high firing rate[33]. Collectively, we can infer that engram cells have several distinct properties that govern the encoding and consolidation of ongoing events[12].

Our data show that the population activity of engram cells is stable and consistent across both NREM and REM sleep sessions and the retrieval session; however, this activity pattern changes upon exposure to a distinct context. This result suggests that the population activity seen during the encoding session is repeated even during post-learning sleep and retrieval sessions to promote the process of memory consolidation. Recent studies have demonstrated that manipulation of post-learning activity modulates the memory consolidation process[13]. Our results support the idea that post-learning replay activity in engram cells closely correlates with memory consolidation.

Using NMF analysis, we were able to obtain several ensembles of neurons coactivated within a certain time window. The synchronous activity shown by engram cells corresponding to a single event does not come from one large set of neurons, but, rather, comprises activity from several sub-ensembles. Each engram sub-ensemble behaves differently over subsequent sessions, which may be attributed to their different synaptic maps[34].

Together, engram sub-ensembles show high reactivation in the subsequent post-learning sleep and retrieval sessions. However, the MS was lower during NREM2, recorded several minutes to 1 h after NREM1, and was comparable to that obtained from shuffled engram data, which suggests that the initial NREM cycle contains a better representation of the recent waking experience. Notably, most engram sub-ensembles that are reactivated during either NREM or REM sleep are reactivated during the retrieval session (aligned in Supplementary Fig. 5A). We detected a group of engram sub-ensembles reactivated during retrieval but not sleep (Fig. 6). This may be due to underdetection of sub-ensembles during sleep because we imaged only the first 1 min of each NREM and REM sleep cycle to avoid photobleaching, even though the total sleep time until the end of the first REM cycle is usually ~2 h. Thus, some of the sub-ensemble replay likely occurred during the non-imaged period of sleep. By contrast, non-engram ensembles that do not replay during sleep are not reactivated in the retrieval session (isolated in Supplementary Fig. 5B) and comprise the vast majority of non-engram ensembles.

The activity of a part of non-engram cells was found to encode place information, while engram cells serve as contextual indexing[32]. We hypothesize that sub-ensembles of engram cells may represent different pieces of information; some sub-ensembles being specific to the learning session, while others being shared between original learning context (sessions A, E) and different context (session F). Shared sub-ensembles may represent an environmental commonality between sessions such as experimental room and experimenter. This seems to be a characteristic feature of CA1 neurons as half of the engram cells are reactivated in different context[32,35]. Even though having this feature, context-specific ensembles of CA1 engram are bearing high weight information for specific memory recall. This implies that these sub-ensembles could have different projections to different memory components: odour, light, and so on, which are then orchestrated to construct an entire complex cognitive event[11,12]. This suggests a mechanism by which the brain machinery incorporates multiple fragments of information into a single representation.

To date, previous engram studies have attempted to elucidate the mechanisms underlying memory by manipulating the activity of engram cells[3,5,8]. However, the physiological nature of engram cells and their behaviour during learning and through the consolidation process has not been previously addressed. Taken together, the findings reported in this study demonstrate that engram cells possess synchronous activity, formed by several sub-ensembles in the engram population. Only in engram cells does this synchronous activity survive through post-learning sleep sessions that contribute to the consolidation process. The present work sheds light on the relationship between activity and coding principles in learning and memory.

## Methods

**Animals**. All procedures involving animals complied with the guidelines of the National Institutes of Health and were approved by the Animal Care and Use Committee of the University of Toyama. C57BL/6J and ICR mice were purchased from Japan SLC, and c-fos-tTA mice were purchased from the Mutant Mouse Regional Resource Centre (stock number: 031756-MU). Thy1-G-CaMP7-T2A-DsRed2 mice have been described previously[23]. Naive mice were wild-type C57BL/6J. All surgery was conducted on male c-fos-tTA mice or c-fos-tTA × Thy1-G-CaMP7 mice with a C57BL/6J background. After weaning, mice were maintained on food containing 40 mg/kg Dox.

The progeny for the c-fos-tTA and c-fos-tTA × Thy1-G-CaMP7 line were generated by in vitro fertilization of eggs from C57BL/6J mice and embryo transfer techniques; these mice were used for behavioural analysis. Superovulation was induced in female C57BL/6J mice by intraperitoneal injection of 7.5 IU of pregnant mare's serum gonadotropin, followed by 7.5 IU of human chorionic gonadotropin 48 h later. Sixteen hours after the human chorionic gonadotropin injection, females

were euthanized, and the cumulus–oocyte complexes were collected. The complexes were placed into 100 μl drops of human tubal fluid (HTF) medium (ARK Resource Co., Ltd.). Spermatozoa were obtained from c-fos-tTA or c-fos-tTA × Thy1-G-CaMP7 male mice. After the animals were euthanized, both caudae epididymides from each mouse were removed and placed into a 200 μl drop of HTF medium to allow capacitation. The sperm suspension (1 μl) was added to the HTF medium containing the cumulus–oocyte complexes. This liquid medium was placed in a sealed modular incubator chamber containing 5% (v/v) $CO_2$ in air and maintained at 37 °C for 5–6 h. The embryos were then washed three times with modified Whitten's medium medium (ARK Resource Co., Ltd., Japan) and incubated for a further 16 h. ICR female mice were mated with vasectomized males. Female mice with vaginal plugs were used for embryo transfer. Recipients were anaesthetized and incised to expose the oviduct. The oviduct wall was perforated with micro-scissors, and a glass pipette containing the two-cell embryos was inserted into the hole. The two-cell embryos (ten embryos/oviduct) were slowly blown into the oviduct. The oviduct was then returned to the abdomen. After embryo transfer, recipients were housed singly for 19–20 days. Genotyping was performed by PCR of genomic DNA isolated from the tails of the pups[23,24]. Mice were maintained on a 12 h light–dark cycle (lights on at 8:00 a.m.) at 24 ± 3 °C and 55 ± 5% humidity, given food and water ad libitum, and housed with littermates until 1–5 days before surgery.

**Viral vectors**. The pLenti-TRE-EYFP and pLenti-TRE-ChR2-EYFP plasmids have been described previously[5]. The pLenti-TRE-hKiKGR plasmid was constructed by replacing the enhanced green fluorescent protein sequence in the TGB plasmid[36] with the humanized KikGR sequence derived from phKikGR1-S1 (Medical and Biological Laboratories Co., Ltd., Nagoya, Japan) using the *Bam*HI and *Mlu*I sites of the TGB plasmid. The pLenti-TRE-EYFP, pLenti-TRE-ChR2-EYFP, and pLenti-TRE-hKiKGR plasmids were prepared using an EndoFree Plasmid Maxi Kit (Qiagen). The LV was prepared as described previously[5], essentially according to the protocol from the laboratory of K. Deisseroth (http://www.stanford.edu/group/dlab/optogenetics/expression_systems.html). The 293FT cells (Invitrogen) were maintained in maintenance medium (Dulbecco's modified Eagle's medium; Gibco, 11995) containing 10% heat-inactivated fetal bovine serum, 2 mM L-glutamine (Gibco, 25030-149), 0.1 mM MEM non-essential amino acids (Gibco, 11140-050), 500 μg/ml geneticin (Gibco, 10131-035), and 1% 100 × penicillin/streptomycin (Gibco, 15140-148). Twenty micrograms of the pLenti plasmid and 48 μg of ViraPower Packaging Mix (Invitrogen, K4975-00) were combined in 6 ml of Opti-MEM (Invitrogen, 31985). The DNA mixture was mixed gently with 6 ml of Opti-MEM containing 144 μl of Lipofectamine 2000 (Invitrogen, 11668). After 20 min, the transfection mix was added to a 225 cm$^2$ flask (BD Falcon, 353138) containing 95% confluent 293FT cells cultured in 28 ml of maintenance medium without geneticin. After 16 h, spent medium was replaced with 30 ml of the virus production medium UltraCULTURE (Lonza, 12-725F) supplemented with 4 mM L-glutamine, 2 mM GlutaMAX-I (Gibco, 35050-061), 0.1 mM minimum essential medium non-essential amino acids, 1 mM sodium pyruvate (Gibco, 11360-070), and penicillin/streptomycin. Seventy-two hours after transfection, the culture medium was collected, filtered with a Millex-HV (0.45 μm) (Millipore), and transferred to a centrifuge tube (Hitachi, 40PA tube). Three millilitres of 20% sucrose in phosphate-buffered saline (PBS) was added to the bottom of each centrifuge tube; the tubes were then centrifuged in a Hitachi P28S swing rotor for 2 h at 50,000 × g. The pellets of TRE-EYFP-LV and TRE-ChR2-EYFP-LV, and that of TRE-hKiKGR-LV, were resuspended in 25 and 12.5 μl of cold PBS (Gibco, 14190) per tube, respectively, and the virus solution was transferred to an Eppendorf tube and centrifuged at 5000 × g for 5 min. The supernatant was collected as described previously[27]; viral titres were approximately 5 × 10$^9$–1 × 10$^{10}$ IU/ml.

**Stereotactic surgery for optogenetics**. Male c-fos-tTA mice were ~20 weeks old at the time of surgery. Mice were anaesthetized with an intraperitoneal injection of pentobarbital solution (80 mg/kg of body weight) and placed in a stereotactic apparatus (Narishige, Japan). The craniotomy for the hippocampus guide cannula was ~1.0 mm in diameter. Each mouse was bilaterally implanted with a guide cannula composed of a stainless-steel tube (internal diameter, 0.255 mm) and a plastic cannula body (Unique Medical Co., Ltd., Japan). The guide cannula targeted the hippocampal CA1 region (anterior to posterior (AP) 2.0 mm, mid to lateral (ML) ±1.4 mm, dorsal to ventral (DV) 0.5 mm from bregma). Micro-screws were screwed into the skull near bregma and lambda, and the guide cannula were fixed to the skull with dental cement. After implantation of the guide cannula, LV (1 μl/injection site) was injected bilaterally through the injection cannula, which was connected to two Hamilton micro-syringes via polyethylene tubes filled with water. The injection cannulae (Unique Medical Co., Ltd.) were inserted into the guide cannulae and targeted CA1 (AP 2.0 mm, ML ±1.4 mm, DV 1.5 mm from bregma). The injection speed (0.1 μl/min) was controlled with a micro-syringe pump (CMA 400, Harvard Apparatus). The injection cannulae were left in place for 10 min and then slowly withdrawn. Dummy cannulae (Unique Medical Co., Ltd.) inserted into the guide cannulae served as protective covers.

For placement of the optical fibre units, mice were anaesthetized with 2.0% isoflurane and the dummy cannulae were removed from the guide cannulae. The optical fibre unit comprised a plastic cannula body and a two-branch-type fibre-optic unit with an optical fibre diameter of 0.25 mm (COME2-DF2-250, Lucir,

Japan). The optical fibre unit was inserted into the guide cannula, and the bodies of the guide cannula and the optical fibre unit were tightly connected with adhesive tape. The tips of the optical fibres targeted CA1 bilaterally (AP 2.0 mm, ML ±1.4 mm, DV 1.0 mm from bregma). Once attached to the mouse, the fibre unit was connected to an optical swivel (COME2-UFC, Lucir, Japan), which was in turn connected to a laser (200 mW, 473 nm, COME-LB473/200, Lucir, Japan) via a main optical fibre. Delivery of light pulses was controlled by a schedule stimulator (COME2-SPG-2, Lucir, Japan) in time-lapse mode.

**Behavioural analysis with optogenetics.** C-fos-tTA mice injected with TRE-EYFP-LV or TRE-ChR2-EYFP-LV were monitored for at least 2 weeks after LV injection. Mice were housed individually in a micro-isolation rack system (FRP BIO2000, CLEA Japan) that comprised 16 individually ventilated boxes (two cages/box) with glass-fibre filters. All training and testing was conducted during the light cycle. The mice were habituated to optical fibre attachment and the shining light for 2 days under ON Dox conditions as follows: mice were anaesthetized with 2.0% isoflurane for 2–3 min. Optical fibres (targeted to CA1) were attached bilaterally through the guide cannulae, and the mice were returned to their home cages for 20–25 min. Optical stimulation (473 nm light, 4 Hz, 15 ms, approximately 7 mW output from the fibre tip)[8] was delivered in a new cage for ~3 min. Five minutes after the end of the stimulation, mice were anaesthetized with approximately 2.0% iso-flurane for about 3 min, the optical fibre unit was detached and the mice were returned to their home cages. Context A had a Plexiglass front, grey sides and back walls (width × depth × height: $175 × 165 × 300$ mm$^3$), and the chamber floor comprised 26 stainless-steel rods (diameter, 2 mm) placed 5 mm apart. The rods were connected to a shock generator via a cable harness. Context B was a cylindrical chamber (diameter × height: $180 × 230$ mm$^2$, respectively) with a white acrylic floor and walls covered with black tape. Contexts A and B were placed in different locations in the same experimental room. The mice were taken off Dox for 2 days and then trained in contextual fear conditioning in the paired or unpaired paradigm of the CPFE procedure[5,37]. During this procedure, mice associate information about the pre-exposure context with an IS experience delivered later if the IS is delivered in the same context (paired), but not if the shock is delivered in a different context (unpaired). The paired paradigm used the same context (context A) for the context pre-exposure and the IS, whereas the unpaired paradigm used distinct contexts: context B for the context pre-exposure and context A for the IS. The memory tests for the paired and unpaired paradigms were performed in contexts A and B, respectively. The experimental room for behavioural analyses adjoined the animal housing room. Animals in their home cages were placed on a desk in the animal housing room for approximately 10 min before each session. Mice were transferred and placed in context A (paired) or B (unpaired) for 6 min (context pre-exposure) and then returned to their home cages. Food pellets without Dox were replaced by pellets containing 1000 mg/kg Dox. One day later, mice were given a 0.8 mA foot shock for 2 s in context A 5 s after acclimation (IS), and then immediately returned to their home cages. After 3–5 h, the mice were returned to the context that was used for context pre-exposure for 3 min to test their fear memory (Test 1). After ~1 h, the mice were anaesthetized with approximately 2.0% isoflurane for about 4 min. Optical fibres were attached through the guide cannulae, and the mice were returned to their home cages for 20–25 min. In Test 2, mice were placed in a new cage for 2 min (laser OFF), then optical stimulation (473 nm light, 20 Hz, 10 ms, approximately 7 mW output from the fibre tip) was delivered to CA1 bilaterally for 2 min (laser ON).

**Immunohistochemistry.** Mice were deeply anaesthetized with an overdose of pentobarbital solution and perfused transcardially with PBS (pH 7.4), followed by 4% paraformaldehyde in PBS. The brains were removed and further post-fixed by immersion in 4% paraformaldehyde in PBS for 24 h at 4 °C. Each brain was equilibrated in 25% sucrose in PBS and then frozen in dry-ice powder. For EYFP staining, coronal sections 50 μm thick were cut on a cryostat and transferred to 12-well cell-culture plates (Corning, Corning, NY, USA) containing PBS. After washing with PBS, the floating sections were treated with blocking buffer (5% normal donkey serum (S30, Chemicon) in PBST) at room temperature (RT) for 1 h. Reactions with primary antibodies were performed in blocking buffer containing rabbit anti-GFP (1:1000, Molecular Probes) antibody at 4 °C for 1–2 nights. After three 10 min washes with PBS, the sections were incubated with a donkey anti-rabbit IgG-AlexaFluor 488 secondary antibody (Molecular Probes, A21206 and A11056) at RT for 3 h. The sections were treated with 4′,6-diamidino-2-phenylindole (DAPI) (1 μg/ml, Roche Diagnostics, 10236276001) and then washed three times (10 min/wash) with PBS. The sections were mounted on slide glass with ProLong Gold antifade reagents (Invitrogen). Images were acquired using a Zeiss LSM 700 confocal microscope with a Plan-Apochromat ×20, 0.8 numerical aperture, objective lens. The photomultiplier tube assignments and pinhole sizes were kept constant. To count the number of EYFP-positive cells, images of the ROI were acquired by collecting z-stacks (~4 optical sections, 3 μm thick). EYFP-positive cells and DAPI-positive nuclei in the ROI were counted by an observer blinded to the experimental conditions.

**Stereotactic surgery for Ca$^{2+}$ imaging and EEG/EMG recording.** All surgery was conducted on male c-fos-tTA × Thy1-G-CaMP7 mice (aged around 12–20 weeks) with a C57BL/6J background. Mice were anaesthetized by

intraperitoneal injection of pentobarbital solution (80 mg/kg of body weight) and placed in a stereotactic apparatus (Narishige, Japan). The craniotomy for LV injection was ~1.0 mm in diameter. TRE-hKiKGR-LV (1 μl/injection site) was injected through an injection cannula connected to a Hamilton micro-syringe via a polyethylene tube filled with water. The injection cannula (Unique Medical Co., Ltd., Japan) was targeted to the right CA1 (AP 2.0 mm, ML 1.4 mm, DV 1.5 mm from bregma). The injection speed (0.1 μl/min) was controlled with a micro-syringe pump (CMA 400, Harvard Apparatus). The injection cannula was left in place for 10 min and then slowly withdrawn. The craniotomy was closed with dental cement, and the skin was closed.

Around 3 weeks after LV injection, complete recovery of mice from the LV injection was confirmed and surgery was performed to implant a gradient refractive index (GRIN) relay lens over the hippocampus, essentially as reported previously[17,19,38]. Mice were anaesthetized by intraperitoneal injection of pentobarbital solution (80 mg/kg of body weight) and placed in a stereotactic apparatus (Narishige, Japan). A 2.0 mm diameter craniotomy was made in the skull to allow access for the cannula lens sleeve (1.8 mm outer diameter and 3.6 mm in length; Inscopix, Palo Alto, CA, USA). A cylindrical column of neocortex and corpus callosum above the alveus covering the dorsal hippocampus was aspirated with saline using a 27-gauge blunt drawing-up needle. The cannula lens sleeve was gently placed on the alveus and fixed to the edge of the burr hole with bone wax melted by low-temperature cautery. The cannula lens sleeve targeted the right hemisphere (AP, 2.0 mm, ML, 1.5 mm) at the centre. Two anchor screws linked to a recording connector were fixed to the frontal part of the skull and served as electrodes for recording the cortical EEG. Stainless wire electrodes were inserted into the neck muscle to record the EMG. A second pair of screw electrodes was fixed to the lateral edge of the skull as a body earth for noise reduction. After positioning additional anchor screws in the skull, the skull and electrodes were covered with dental cement to fix the cannula lens sleeve to the skull and anchor screws.

More than 2 weeks after the surgery to implant the cannula lens sleeve and EEG/EMG electrodes, mice were anaesthetized with pentobarbital solution (80 mg/kg of body weight; intraperitoneal injection) and a GRIN lens (1.0 mm outer diameter and 4.0 mm length; Inscopix, CA, USA) was inserted into the cannula lens sleeve and fixed in place with ultraviolet-curing adhesive (Norland, NOA 81). An integrated miniature microscope (nVista HD, Inscopix, CA, USA)[19] with a microscope baseplate (Inscopix, CA, USA) was placed above the GRIN lens, allowing observation of G-CaMP7 fluorescence and blood vessels in CA1. The microscope baseplate was fixed to the head of an anchor screw using dental cement by which the GRIN lens was shaded; then, the integrated microscope was detached from the baseplate. The GRIN lens was covered by attaching the microscope baseplate cover to the baseplate until it was time for Ca$^{2+}$ imaging.

**Microscopy and EEG/EMG recording of freely moving mice.** Calcium imaging was performed during the light cycle, and Ca$^{2+}$ events were captured with nVista acquisition software (Inscopix, CA, USA) at 20 frames/s with the CMOS sensor at maximum gain and nVista HD light-emitting diode (LED) power at 60%. Digitized EEG/EMG traces, at a sampling rate of 1 kHz with bandpass filtering (low cut-off frequency of 0.5 Hz and high cut-off frequency of 300 Hz for EEG; low cut-off frequency of 5 Hz and high cut-off frequency of 3000 Hz for EMG), were amplified and collected using Unique Acquisition (Unique Medical). Fast Fourier transform was used to score sleep/wakefulness states manually based on EEG/EMG signals on all consecutive 16 s epochs, according to the following criteria: wakefulness—high EMG and low EEG voltage with high-frequency components; NREM sleep—low EMG and high EEG voltage with high $\delta$ (0.5–4 Hz) frequency components; and REM sleep—low EMG indicating muscle atonia and low EEG voltage with high $\theta$ (6–9 Hz) frequency[39]. Contexts A and B (used in the behavioural analysis with optogenetics) were placed in close, but different, locations within the same experimental room. As before, context A had a Plexiglass front, grey sides and back walls (width × depth × height: $175 × 165 × 300$ mm$^3$) and the chamber floors comprised 26 stainless-steel rods (diameter, 2 mm) placed 5 mm apart. The rods were connected to a shock generator via a cable harness. Context B was a cylindrical chamber (diameter × height: $180 × 230$ mm$^2$, respectively) with a white acrylic floor and walls covered with black tape. c-fos-tTA × Thy1-G-CaMP7 mice injected with TRE-KikGR-LV were moved to a micro-isolation box (FRP BIO2000, CLEA Japan) located in a Faraday cage (Narishige, Japan) for maintenance during experiments. For habituation to the nVista HD attachment and to the cables for EEG/EMG recording, mice were subjected to test trials of Ca$^{2+}$ imaging and EEG/EMG recording for around 20 min/day for 3 days. The day after completion of habituation, mice were taken off Dox. Two days later, mice were anaesthetized for about 3 min with approximately 2.0% isoflurane. To decrease the green fluorescence from KikGR expressed before contextual learning, two 15 s pulses of 365 nm light (1.5 min interval between pulses) were delivered to the GRIN lens (and hence to CA1) from a LED light source (BL-LED-365, OPTO-LINE, Inc., Japan) via an optic fibre (SH4001 Super Eska, NA: 0.5, Mitsubishi Rayon Co., Ltd., Japan). Then, a snapshot of the CA1 field with decreased KikGR green fluorescence before learning was taken through the nVista HD miniature microscope using the acquisition software (Inscopix, CA, USA). After 1.5–2 h, mice were introduced to a novel context (context A) for 6 min under Ca$^{2+}$ imaging (session A). Mice, with the nVista HD still attached, were returned immediately to their home cage and

connected to the EEG/EMG recording system. Recording corresponding to NREM1 (session B) was carried out close to the timing of onset of the first sleep after learning. NREM2 (session C) was separated from NREM1 by about 10–50 min. REM (session D) was defined as the first period of REM sleep that lasted for at least 1 min. The nVista LED was turned off for the post-learning sleep period except during actual imaging. To avoid photobleaching, imaging was performed only for the first 1 min of each NREM or REM sleep cycle. After waking from a period of sleep that included a REM recording, mice were disconnected from the nVista HD and EEG/EMG recording system. One day later, mice were again anaesthetized for about 3 min with 2.0% isoflurane. A snapshot of the CA1 field showing expression of KikGR green fluorescence was taken by the nVista HD acquisition software (Inscopix, CA, USA). Then, photoconversion of KikGR fluorescence before memory retrieval was induced as described above: two 15 s pulses of 365 nm light separated by a 1.5 min interval. After 1.5–2 h, the mice were introduced into context A for 3 min under $Ca^{2+}$ imaging (session E). The mice, with the nVista HD microscope attached, were returned immediately to their home cage. Then, after 1.5–2 h, the mice were introduced into context B for 3 min under $Ca^{2+}$ imaging (session F).

**$Ca^{2+}$ imaging data acquisition, processing, and cell sorting.** Calcium transients were captured (20 frames/s) using the nVista acquisition software (Inscopix, CA, USA). The captured images were processed by the Inscopix Mosaic software, as described previously[40]. Initially, each session movie was spatially down-sampled and smoothed by a factor of 2, for motion artefacts. Motion correction was performed (correction type: translation only, spatial mean ($r = 20$ pixels) subtracted, and spatial mean applied ($r = 5$ pixels)), with blood vessels used as a landmark to maintain the same field of view (FOV) and correct for the motion artefacts. Next, the movie was processed by the ImageJ software (NIH). Each session movie was divided (pixel by pixel) by a low-passed ($r = 20$ pixels) filtered version. For tracking cells across multiple imaging session days, single session movies were concatenated into one total movie containing all the recording sessions. Then, motion correction was employed on the total movie using a single frame as a reference frame to ensure that XY translation was adjusted across all recording sessions (Fig. 2c), similar to a previous report[41]. To assure the accuracy of the motion correction, the activity of several cells in the total movie was tracked across 2 days to ensure that cells did not shift their spatial location during the movie (Fig. 2d and Supplementary Fig. 1D). Subsequently, the $\Delta F(t)/F0 = (F(t) - F0)/F0$ value was calculated for each movie session (where $F0$ is the mean image obtained by averaging the entire movie for that session) to measure changes in fluorescence. Finally, all movie sessions were concatenated in Mosaic software to obtain the movie for the entire behavioural session. Once the $Ca^{2+}$ movie was processed, a fully automated $Ca^{2+}$ sorting algorithm called the HOTARU (high-performance optimizer for spike timing and cell location via linear impulse) system[23,27] was used to identify and extract active cells. The HOTARU system is highly accurate at estimating calcium activity; the activity of overlapping cells can be reconstructed efficiently by removing crosstalk from the observed signals[27]. The algorithm was provided with information about the kinetics of G-CaMP7 and then allowed to run through the $Ca^{2+}$ movie to detect the locations of local fluorescent maxima that met a minimum $\Delta F(t)/F0$ criterion.

**Identifying engram cells through KikGR expression.** To identify KikGR-expressing cells, nVista acquisition software (Inscopix, CA, USA) took a snapshot of CA1 under isoflurane anaesthesia 1 day after novel context exposure. Using the ImageJ software (NIH), the snapshot was spatially down-sampled and smoothed by a factor of 2, then manually re-aligned to the same FOV, using blood vessels as a landmark, as the captured $Ca^{2+}$ movie. Next, the snapshot was divided (pixel by pixel) by a low-passed ($r = 20$ pixels) filtered version. Then, a mask was applied to the snapshot, encompassing the area containing gene expression (Fig. 2d and Supplementary Fig. 1B). Later, a cutting threshold (92% average) was applied to remove the background and maintain the top 8% signal intensity. The resulting image was then binarized (Binary options: Iteration, 1; Count, 2; Close (fill small holes between pixels); and Watershed (auto-separation of several combined particles)). Finally, the particles in the output image were analysed automatically by ImageJ; the detected ROIs were considered to be the locations of gene-expressing cells. As mentioned previously, the automated sorting system generated spatial filters for the locations of detected active calcium transients. ImageJ-detected ROIs represent the locations of KikGR cells. Subsequently, the positions of all cells detected within the ROI were compared. Some ROIs for gene-expressing cells did not fully overlap with the detected calcium transients, and other ROIs were clearly not neurons; both were discarded (see also Supplementary Fig. 1C). Only ROIs that, through visual inspection, showed complete matches with active cells were identified as engram cells. Every identified engram cell was verified by researchers via visual scrutiny.

**Mathematical analysis.** *Correlation matrix overlaps*: Temporal correlation between pairs of engram cells or non-engram cells was calculated using a 1 s sliding time window at time steps of 200 ms. The correlation matrix calculated at time $t$ is denoted by $\hat{C}(t) \equiv \left( C_{ij}^t \right)$, where $i$ and $j$ are neuron indices and $t$ refers to the midpoint of the time window. The similarity between the two correlation matrices

at $t$ and $t'$ was calculated as $M(t, t') \equiv \frac{1}{N(N-1)} \sum_{i \neq j} C_{ij}^t C_{ij}^{t'}$[29], where $N$ is the number of cells in the dataset. To compare engram, non-engram and shuffled engram cells, the MS with respect to $t'$ was calculated as $M_{\text{total}}(t) \equiv \sum_{t'} M(t, t')$, which measures the degree to which the correlation pattern of population activity at $t$ is repeated at different times within the whole dataset. The larger the value, the more often the correlation pattern looks similar to that obtained at $t$. To check the significance of the results, Monte Carlo resampling was applied as a shuffling control. In the resampling, the correlation matrix overlaps were calculated in 10,000 individual shuffling samples to generate a converged distribution to assess significance. *Population vector distance*: PVD quantifies the difference in calcium signals obtained from a group of neurons during two distinct sessions. Calcium signals from two distinct sessions, X and Y, of the same neuron group were compared using the restricted Mahalanobis distance, which is defined on the basis of the Mahalanobis distance[42]. The restricted Mahalanobis distance is defined as

$$PVD(X, Y) \equiv \sqrt{\max_\varphi \sum_{i=1}^{10} \left[ v_{\varphi(i)}^T (\mu_X - \mu_Y) \right] / \lambda_{\varphi(i)}},$$ where $\mu_X$ and $\mu_Y$ are the means of X and Y, respectively, $\lambda_{\phi(i)}$s are eigenvalues of the covariance matrix of the union of X and Y, $v_{\phi(i)}$ is the corresponding eigenvector and $\phi$ is a bijective mapping reordering indices of eigenvalues and eigenvectors. Intuitively, this equation calculates the maximum distance within a restricted number of dimensions. This definition enables us to compare the PVD across different groups of neurons, that is, engram cells and non-engram cells, that contain different numbers of neurons. *Non-negative matrix factorization*: NMF was used to extract population activity patterns from the calcium signals dataset. More specifically, NMF finds an optimal factorization of the data matrix $\hat{D}$, with a pattern matrix $\hat{B}$ and the corresponding intensity matrix $\hat{C}$, that is, $\hat{D} \approx \hat{B}\hat{C}$ (Supplementary Fig. 3A). Here, the rows of $\hat{D}$ represent the time series of the signals from individual neurons, each column vector of $\hat{B}$ represents a synchronously activated neuron ensemble (population activity pattern) and each row of $\hat{C}$ represents the time series of the activation intensity of the corresponding pattern. To search for such a factorization, the cost function defined by $E \equiv \sum_{ij} \left( D_{ij} - \sum_k B_{ik} C_{kj} \right)^2$ was minimized using both multiplicative and additive algorithms[30]. Random initial entries from matrices $\hat{B}$ and $\hat{C}$ were used for 1000 attempts at minimization, and the pair of pattern and intensity matrices minimizing the cost function were chosen to be the best factorization. The cost function becomes smaller if more patterns are introduced, but the model (i.e. the right-hand side of the cost function) becomes more complex. This trade-off between cost minimization and model complexity was optimized using the AICc[31] to determine the optimal number of patterns (i.e. the number of columns in the pattern matrix). In Supplementary Fig. 3A, the original data were shuffled independently in terms of both the neuron indices and the timestamps to obtain a shuffled dataset. Ten shuffled datasets were constructed for each sample in each session. Because the ratio of the standard deviation to the mean is small (~0.03), this number of samples should be sufficient. *Matching score*: The overall similarity between pattern vectors in sessions X and Y was measured according to the normalized dot product $\vec{v}_i^X \cdot \vec{v}_j^Y$ for all possible pattern pairs across the two sessions, noting that the dot product is equivalent to the cosine of the angle between the pattern vectors[43,44]. To this end, the MS between sessions X and Y was defined as

$$MS(X, Y) \equiv \frac{1}{N_X} \sum_{i \in X} \Theta \left[ \sum_{j \in Y} \Theta \left( \vec{v}_i^X \cdot \vec{v}_j^Y - c \right) - d \right],$$ where $\vec{v}_i^X$ ($\vec{v}_j^Y$) is the $i$th ($j$th) pattern vector in session X (Y), $N_X$ is the number of patterns in session X and $\Theta(\cdot)$ is a step function. The constant $d$ is an arbitrary positive number smaller than unity. This scoring function yields the portion of patterns in session X that have a normalized dot product larger than $c$ with any of the patterns in session Y. A threshold of $c = 0.6$ was used throughout the study (Supplementary Fig. 5B). Monte Carlo resampling was conducted for the MSs of engram cells. In the resampling, MSs for datasets with shuffled timestamps and neuronal indices are calculated to assess the significance. For each session-pair comparison, the mean of the MS for the shuffled data is compared with that calculated from the original data. To calculate the mean of the MS for the shuffled data, 40 shuffled samples were used. Since the resampling is not for the purpose of constructing a random-number distribution, the calculation of the mean value converged, and the number of samples is sufficient. Normalized data was calculated by subtracting shuffled data of engram and non-engram cells from their corresponding MS data.

**Statistical analysis.** Statistical analyses were performed in GraphPad Prism 6 (GraphPad Software) and Excel (Microsoft), with Statcel 3 (OMS, Japan). Data from two groups were compared with Student's $t$-test. Comparison of data between three groups was constructed using a one-way analysis of variance (ANOVA). Comparison of data between two grouped at different time points were constructed using a two-way repeated-measures ANOVA followed by Bonferroni's multiple comparisons test. Comparison of data between two paired groups was performed using a paired $t$ test or the Wilcoxon's signed-rank test (no assumption that data were normally distributed). One-tailed comparisons were used whenever the difference between the two groups was expected to be in a single direction. Two-tailed comparisons were used whenever the difference between the two

groups was expected to be in either direction. Quantitative data are expressed as the mean ± s.e.m.

**Reporting summary**. Further information on research design is available in the Nature Research Reporting Summary linked to this article.

## Data availability
The materials and code that support the findings of this study are available from the corresponding author upon reasonable request. Custom-made program for NMF analysis can be found in https://github.com/fccaa/NMF_custom.

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

## Acknowledgements
We thank S. Tsujimura for genotyping the transgenic mice, M. Nomoto for introducing the CNMF-E and all members of the Inokuchi laboratory at the University of Toyama and the Fukai laboratory at RIKEN CBS for support and valuable discussion. This work was supported by the JST PRESTO program JPMJPR1684 (N.O.) and the JST CREST program JPMJCR13W1 (K.I.), by the Japan Society for the Promotion of Science KAKENHI (Grant Numbers JP26640008 and JP16H04653 (N.O.), JP16H06276 (M.Os.), JP15H05723 (J.N.), JP18H02595 (T.S.), and JP23220009 and JP18H05213 (K.I.)), by a Grant-in-Aid for Scientific Research on Innovative Areas "Memory dynamism" (JP25115002 (K.I.) and JP26115504 (J.N.)), by "Willdynamics" (JP16H06401 (T.S.)), by "Artificial Intelligence and Brain Science" (JP17H06036 (T.F.)) from the Ministry of Education, Culture, Sports, Science and Technology (MEXT), by the Rotary Yoneyama memorial foundation (K.G.), by the Lotte Research Promotion Grant (N.O.), by the Naito Foundation (N.O.), by the Ichiro Kanehara Foundation (N.O.), by the Tamura Science and Technology Foundation (N.O.), by the Mitsubishi Foundation (K.I.), by the Uehara Memorial Foundation (K.G. and K.I.) and by the Takeda Science Foundation (support to N.O. and K.I.).

## Author contributions
N.O. and K.I. supervised the entire project. K.G., N.O., C.C.A.F., T.F. and K.I. designed the study. N.O. cloned the lentiviral construct, prepared the LVs and performed the behavioural experiments and histology. H.N. and M.M. contributed to the generation and maintenance of transgenic animals. K.G., N.O. and Y.S. performed animal surgery. M.S., M.Oh., J.N. and Y.H. established the Thy1-G-CaMP7 transgenic mice. T.Ki contributed to establishing the Ca²⁺ imaging. K.G., H.A., T.Ta, R.O.-S., M.Os., and T.Ki contributed to the analysis of the Ca²⁺ imaging data. C.C.A.F. and T.F. performed the mathematical analyses. S.S. and T.S. contributed to sleep-stage discrimination. K.G., N.O., C.C.A.F., T.F. and K.I. wrote the manuscript.

## Additional information

**Competing interests:** Y.H. receives research funding from Fujitsu Laboratories Ltd. and Dwango. The other authors declare no competing interests.

