## [Transparent Peer Review File · Nature Communications]

Editorial Note: Parts of this Peer Review File have been redacted as indicated as we could not obtain permission to publish the reports of Reviewer #3 for revised versions of this manuscript.

Reviewers' comments:

Reviewer #1 (Remarks to the Author):

Ghandour et. al invented a novel method to track a pre-labeled cells with miniaturized endoscope without interrupting GCaMP signals by utilizing photoconvertable fluorescent protein KikGR. They showed context engram cells maintain their network with subensembles to store information of the context. They tried to prove the network mechanism of memory storage. They demonstrated that context specific engram maintains their network with similar network activity with specific subensembles. They also demonstrated that population activity of engram is quite stable during NREM and REM sleep sessions and retrieval session although this activity pattern changes upon to a distinct context. This ms is interesting and suitable for publication. I have only minor concerns that could be considered to improve the ms.

1. The authors used DOX-off Tet-tag system to label "engram cells". Considering relatively long period of DOX-off, I am afraid that the labeled cells by this way may contain a significant proportion of non-engram cells. In figure 2, dox-off period is very long including learning session and even sleep session. Therefore, there is certainly a possibility that "subensemble cells" could be a real ensemble cells. The authors should discuss it.

2. Strictly speaking, the labeled neurons are not context + shock specific engrams. Rather, they are context specific engrams labeled during pre-exposure period. This could be a major weakness of this ms as the so-called 'engram cells' they suggest are not the engram formed during fear memory acquisition. Even though the reactivation of 'context engram cells' may cause the recall of fear memory, it is possible that the context engram population contain false-positive cells and false-negative cells of fear memory engram. This point should also be discussed.

3. Within a single engram population, the authors detected many subensembles. Some subensembles were more likely to be reactivated during sleep and retrieval. Although the authors hypothesized that such sub-ensembles may represent distinct sets of information, the synaptic nature of these subensembles may be discussed further. The synaptic connections among cells within a specific subensemble may be strengthened specifically and this can be demonstrated by a recent dual eGRASP technology developed to analyze the synaptic changes between engram cells (Choi et al. 2018, Science). The authors may also include a reference of CA1 engram activation (Roy et al. 2015, Science).

Reviewer #2 (Remarks to the Author):

The authors of this manuscript investigated spatial memory coding in hippocampal engram cells using a head-mounted calcium imaging approach. They used a novel KikGR (Kikume Green Red) system to tag neurons active ("engram cells") during spatial exploration. They then imaged the calcium activity of tagged and untagged neurons during memory encoding, sleep and retrieval sessions. Their main novel finding is the existence of engram "sub-ensembles". If a sub-ensemble is reactivated during sleep, it is more likely to be active during memory retrieval.

This paper combines an engram tagging study with a calcium imaging study. This is a fairly novel approach. However, I have serious reservations about the experimental design and data analyses used to support their main conclusions:

- I am concerned about the authors' pipeline for processing calcium videos between experimental sessions (Fig. 2C). Their methods do not seem to adequately account for possible registration errors. These errors have the potential to entirely skew their results and interpretation. The authors use a motion-correction based approach on their raw videos to establish whether they are imaging the same cells across sessions. However, aligning the raw videos is not the appropriate approach. Instead, the authors should use the spatial footprints extracted from their HOTARU

algorithm as the basis for registration. They should align these footprints between sessions to ensure they are reliably tracking cells across days. Without this, their conclusions are not supported by any data.

- I am concerned about the authors' statistical analyses in Fig. 4B/C and Fig. 5F-I. These do not seem to be the best way to answer the questions that they pose. In these cases, the authors seem to want to compare differences between their engram and non-engram groups across time. However, it appears that their data do not satisfy the necessary conditions for a parametric test, so they have opted for a series of Wilcoxon signed-rank tests. In order to do this, they have had to construct difference scores (Fig. 4D, 5G/I), and repeatedly compared test session A to subsequent sessions. Of course, this approach needlessly introduces potential for error and is overly complicated. It is unclear why the authors did not use a more appropriate Growth Curve Analysis (GCA) to parse the differences between their groups across time. This analysis would allow them to directly compare the differences between their groups and at different timepoints without having to generate additional difference figures as they have done here. GCA does not assume linearity, which may be important for the authors' data.

- In Fig. 5F, the authors need to normalize to a shuffle control (a principle similar to the one they use in Fig. 3C/D). They are making a comparison between engram and non-engram matching scores, but the differences they see could be entirely driven by the higher activity of engram cells (a finding they report). Normalizing the engram and non-engram matching scores to their shuffled distributions, would allow them to make valid comparisons.

- The details of the sleep imaging experiments are not sufficient. For example, it is left as an open question whether the authors continuously imaged during the sleep sessions, or if only portions of the sleep period were sampled. If the authors used continuous imaging, then how do they control for photobleaching? If they did not, how did they determine precisely when to image?

- Further to the above point, I think the authors must extend their analyses in Fig. 6A/B to include all sessions, including exposure to the different context. Without this important control, there is no justification to ensure that these ensembles represent a specific memory. These ensembles should only be active in the context of initial learning, and should not be active in session F.

- Furthermore, the authors provide very little evidence that their task has induced learning (or a memory) at all. Without some sort of behavioral correlate of memory, it is unclear whether exposure to a context is sufficient to induce memory. And it is unclear whether active labelled cells represent anything at all.

- I am concerned that supplementary Fig. 3B/C is not sufficiently stringent to provide convincing evidence of the authors' point. The authors claim that extracted patterns are composed of activity of constituent neurons. This comparison should be done using summary data from all patterns and their constituents, rather than simply providing one representative example for each case.

Reviewer #3 (Remarks to the Author):

General comments

In this paper, Ghandour and colleagues describe how they developed a novel imaging system and how this allowed them to study the activity patterns of engram and non-engram cells in vivo, during learning, subsequent sleep and recall. They show that, compared to non-engram cells, engram cells are more rhythmically activated during learning and that their activity is more stable across memory stages. Among engram cell populations, they identify sub-ensembles characterised by specific patterns of activity. The sub-ensembles that replay during sleep are also reactivated

during retrieval.

This manuscript is interesting because it investigates the specific population dynamics of engram cells. The methods used to address this question are appropriate and sound. To the best of my knowledge this is the first reported use of the photoconvertible protein KiGR combined with calcium imaging to study memory engrams. Overall, the data presented here support the authors' conclusions.

The scientific narrative is well structured and flows naturally from one idea to the next. However, on multiple occasions, comprehension is hindered by grammar mistakes, poor word choices, and awkward sentence structures. Careful editing should allow this issue to be remedied.

In my opinion, the authors should contextualise their findings in the existing engram literature more thoroughly and emphasize what they believe makes this research novel.

Figure 1

The experiments and results presented in this figure and the related text are convincing and in agreement with the established engram literature. The experimenters show that exposure to a context can induce freezing, when this context was previously paired with a shock. Optogenetically activating engram cells labelled in this context also induces freezing. I note that in Fig. 1 a), on the line for the unpaired condition, the word "freezing" is accompanied by an arrow pointing upwards, suggesting that freezing increases in this condition. However, as far as I understand, this is not the case (see Fig. 1 c)).

Figure 2

The authors describe how they labelled and identified engram cells with KikGr and subsequently rendered this signal invisible for Ca²⁺ imaging. This is an elegant method that allows comparison between engram cell and non-engram cell activity in vivo.

Figure 3

Thanks to a correlation matrix analysis, adequately described in the Methods section, this figure shows that engram cells exhibit more repetitive activity than non-engram cells during memory encoding. However, the authors make not attempt at explaining the rhythmic activity that also exists in non-engram cells. Could this be linked to hippocampal rhythms? I also would have liked to this this type of analysis for subsequent memory processing stages.

Figure 4

The result in this figure show that the activity of engram cells, as described by the Population Vector Difference, is more stable than that of non-engram cells. Fig. 4 d also highlights a significant difference between memory retrieval and exposure to a different context, but the text does not reflect on the relevance of that difference.

Figure 5

Non-negative matrix factorization is an interesting method to identify and compare patterns of activity in cell populations. It allowed the authors to show that sub-ensembles of engram cells are characterised by specific patterns of activity that are repeated throughout experimental sessions, but less so when mice are exposed to a new context. In contrast, non-engram cells do not exhibit this tendency.

Figure 6

Tracking activity patterns across sessions showed that engram cells that are reactivated during sleep are more often reactivated during retrieval compared to non-engram cells and that non-engram cells that are activated during learning are usually not reactivated during subsequent sessions.

On top of comparisons between engram and non-engram cells for each group, it would have been advisable to show statistical comparisons between groups (e.g. engram sub-ensembles reactivated during sleep and retrieval vs. during retrieval but not sleep).

We thank you for handling our manuscript and the reviewers for their helpful and constructive comments. We have addressed all of the reviewers' comments, and the point-by-point responses to the comments are described below. In the revised manuscript, all corrections are indicated in red. We hope that our revised manuscript is now suitable for publication in *Nature Communications*.

Reviewer 1

Ghandour et. al invented a novel method to track pre-labelled cells with miniaturized endoscope without interrupting GCaMP signals by utilizing photoconvertable fluorescent protein KikGR. They showed context engram cells maintain their network with subensembles to store information of the context. They tried to prove the network mechanism of memory storage. They demonstrated that context specific engram maintains their network with similar network activity with specific subensembles. They also demonstrated that population activity of engram is quite stable during NREM and REM sleep sessions and retrieval session although this activity pattern changes upon to a distinct context. This ms is interesting and suitable for publication. I have only minor concerns that could be considered to improve the ms.

We appreciate the reviewer's careful reading, positive feedback, and constructive comments, which have truly helped us to increase the quality of the manuscript. We have taken all of the comments into consideration while revising the paper, as detailed below.

The authors used DOX-off Tet-tag system to label "engram cells". Considering relatively long period of DOX-off, I am afraid that the labeled cells by this way may contain a significant proportion of non-engram cells. In figure 2, dox-off period is very long including learning session and even sleep session. Therefore, there is certainly a possibility that "subensemble cells" could be a real ensemble cells. The authors should discuss it.

Several lines of evidence suggest that the DOX-off duration used in our work is appropriate to specifically label the engram cells without expanding the size of the engram population. The duration used for labelling with the lentivirus system is essentially the

same as that used in a previous study from our lab, in which we specifically labelled CA1 engram cells corresponding to a contextual memory (Ohkawa et al., 2015, *Cell Rep*). Optogenetic manipulation of the engram cells, which had been labelled via this protocol, affected the corresponding memory in a specific manner. Furthermore, the labelling efficiency in the present study was 12% on average (Fig. 1e), which is comparable to the CA1 Tet-Tag labelling efficacy reported previously (Tyler et al., 2012, *Curr Biol*; Ohkawa et al., 2015, *Cell Rep*). Moreover, the labelling efficacy of KikGR⁺ in G-CaMP7⁺ cells was 8.09% ± 0.84% (see Supplementary Table 1) overall, which is again comparable to the above-mentioned previously reported results.

Strictly speaking, the labeled neurons are not context + shock specific engrams. Rather, they are context specific engrams labeled during pre-exposure period. This could be a major weakness of this ms as the so-called ‘engram cells’ they suggest are not the engram formed during fear memory acquisition. Even though the reactivation of ‘context engram cells’ may cause the recall of fear memory, it is possible that the context engram population contain false-positive cells and false-negative cells of fear memory engram. This point should also be discussed.

Throughout the study, we focused on the dynamics of context engram cells, not context + shock engram cells. Please note that contextual learning (without shock) was used for the calcium imaging. We employed context + shock only in the experiments illustrated in Fig. 1. The purpose of the experiments in Fig. 1 was to show that the labelled cells indeed act as an engram for the contextual memory. The protocol to label engram cells for contextual memory in Fig. 1 was the same as the protocol used to label engram cells in the other figures. We took advantage of a context pre-exposure and immediate shock paradigm, in which mice associate context A (a pre-exposed context) with a temporally separated shock delivered in the same context (the paired condition). Therefore, we labelled engram cells corresponding to context A, which was only later associated with shock. We showed that the shock memory could be retrieved by activating the labelled cells

(corresponding to the context memory). This result indicates that the cells labelled during context exposure really do encode context memory.

Within a single engram population, the authors detected many subensembles. Some subensembles were more likely to be reactivated during sleep and retrieval. Although the authors hypothesized that such sub-ensembles may represent distinct sets of information, the synaptic nature of these subensembles may be discussed further. The synaptic connections among cells within a specific subensemble may be strengthened specifically and this can be demonstrated by a recent dual eGRASP technology developed to analyze the synaptic changes between engram cells (Choi et al. 2018, *Science*). The authors may also include a reference of CA1 engram activation (Roy et al. 2015, *Science*).

We agree that the dual eGRASP system is useful for investigating the synaptic connections of selected sub-ensembles. As suggested, we have added this point to the Discussion section and have cited Choi et al., 2018, *Science*. Please see lines 9-10, page 14 of the revised manuscript. The new sentence is as follows:

“Each engram sub-ensemble behaves differently over subsequent sessions, which may be attributed to their different synaptic maps”

We have also cited the CA1 engram reactivation study from Dr. Tonegawa’s group (Ryan et al., 2015, *Science*).

Reviewer 2

The authors of this manuscript investigated spatial memory coding in hippocampal engram cells using a head-mounted calcium imaging approach. They used a novel KikGR (Kikume Green Red) system to tag neurons active (“engram cells”) during spatial exploration. They then imaged the calcium activity of tagged and untagged neurons during memory encoding, sleep and retrieval sessions. Their main novel finding is the existence of engram “sub-ensembles”. If a sub-ensemble is reactivated during sleep, it is more likely to be active during memory retrieval.

This paper combines an engram tagging study with a calcium imaging study. This is a fairly novel approach. However, I have serious reservations about the experimental design and data analyses used to support their main conclusions.

We thank the reviewer for the thorough review, which has helped us improve the quality of the manuscript. We have taken all of the comments into consideration while revising the paper, as detailed below.

- I am concerned about the authors' pipeline for processing calcium videos between experimental sessions (Fig. 2C). Their methods do not seem to adequately account for possible registration errors. These errors have the potential to entirely skew their results and interpretation. The authors use a motion-correction based approach on their raw videos to establish whether they are imaging the same cells across sessions. However, aligning the raw videos is not the appropriate approach. Instead, the authors should use the spatial footprints extracted from their HOTARU algorithm as the basis for registration. They should align these footprints between sessions to ensure they are reliably tracking cells across days. Without this, their conclusions are not supported by any data.

Aligning the sessions across days to make sure that we are indeed imaging from the same cells is crucial, as pointed out by the reviewer. In Fig. 2e, we show the spatial footprints, extracted by the HOTARU algorithm, of cells activated on both day 1 and day 2 together with their Ca^{2+} traces and actual fluorescence, with the number of frames indicated. This figure clearly demonstrates that the same cells were recorded across days without spatial shifting in the field of view.

Per the reviewer's request, we have added an extra panel (Supplementary Figure 1E) showing the spatial footprints of the temporally matched cells in the concatenated movie overlaid with the spatial footprints of cells in the single movie(s) (day 1 and/or day 2). This should provide assurance that there was no spatial shift.

- I am concerned about the authors' statistical analyses in Fig. 4B/C and Fig. 5F-I. These do not seem to be the best way to answer the questions that they pose. In these cases, the authors seem to want to compare differences between their engram and non-engram groups across time. However, it appears that their data do not satisfy the necessary conditions for a parametric test, so they have opted for a series of Wilcoxon signed-rank tests. In order to do this, they have had to construct difference scores (Fig. 4D, 5G/I), and repeatedly compared test session A to subsequent sessions. Of course, this approach needlessly introduces potential for error and is overly complicated. It is unclear why the authors did not use a more appropriate Growth Curve Analysis (GCA) to parse the differences between their groups across time. This analysis would allow them to directly compare the differences between their groups and at different timepoints without having to generate additional difference figures as they have done here. GCA does not assume linearity, which may be important for the authors' data.

The reviewer requested that we consider the effect of time on the differences between the engram and non-engram groups, especially the effect on population vector distances and matching scores (Fig. 4c and Fig. 5f, h). For this purpose, he/she recommended that we perform GCA. We discussed this but decided to use a two-way repeated measures ANOVA. This type of analysis is commonly used in the field of neuroscience to assess the effect of time on differences in the features of two groups. The results clearly indicate significant differences in engram versus non-engram or shuffled engram data. These results are shown in Fig. 4c and Fig. 5f, h of the revised version. In addition, we performed GCA, as suggested, and the results supported the same conclusion. The GCA report is attached to the end of this letter, but has not been added to the revised manuscript.

We also performed paired *t*-tests to evaluate the differences in engram versus non-engram or shuffled engram data in each session. The results from the Wilcoxon signed-rank tests in the previous version of manuscript have been replaced by those from the paired *t*-test (Fig. 4d and Fig. 5g, i in the revised version).

- In Fig. 5F, the authors need to normalize to a shuffle control (a principle similar to the one they use in Fig. 3C/D). They are making a comparison between engram and non-engram matching scores, but the differences they see could be entirely driven by the higher activity of engram cells (a finding they report). Normalizing the engram and non-engram matching scores to their shuffled distributions, would allow them to make valid comparisons.

The reviewer recommended that we normalize the original data to the shuffled data prior to performing the matching score analysis because of the possibility that the high matching scores of the engram cells may be due to the higher activity in these cells. However, our goal in this study was to extract the physiological differences between engram and non-engram cells, and higher activity is a prominent feature of the engram cells. To demonstrate that the difference between engram and non-engram cells shown in Fig. 5f is not solely due to the higher activity in engram cells, we compared the original engram-cell data with shuffled engram-cell data in the original manuscript (Fig 5h, i). The result clearly indicates that the high matching score in engram cells is derived from their coordinated activity because this coordination is destroyed by shuffling.

- The details of the sleep imaging experiments are not sufficient. For example, it is left as an open question whether the authors continuously imaged during the sleep sessions, or if only portions of the sleep period were sampled. If the authors used continuous imaging, then how do they control for photobleaching? If they did not, how did they determine precisely when to image?

We agree with the reviewer that our description of the imaging procedure during sleep was not sufficiently detailed in the previous version of the manuscript. We did not image continuously throughout the entire sleep session; hence we are certain that there was no photobleaching. We monitored the EEG/EMG signals continuously to identify cycles of NREM and REM sleep. We imaged the first 1 min of each NREM and REM cycle. Therefore, the total duration of sleep imaging during the 2 h post-learning period was less than 5 min. The nVista LED was always turned off except during the actual imaging. We

have added more detail to the description of sleep imaging in the Methods section, lines 10–12, page 23, as follows:

“The nVista LED was turned off for the post-learning sleep period except during actual imaging. To avoid photobleaching, imaging was performed only for the first 1 min of each NREM or REM sleep cycle.”

Related to this point, we have also added a statement regarding underdetection of sub-ensembles during sleep. Please see our reply to Reviewer 3, Fig. 6.

- Further to the above point, I think the authors must extend their analyses in Fig. 6A/B to include all sessions, including exposure to the different context. Without this important control, there is no justification to ensure that these ensembles represent a specific memory. These ensembles should only be active in the context of initial learning, and should not be active in session F.

As suggested, we have extended the analyses in Fig. 6A/B to include session F. The results indicate that half of the sub-ensembles from engram cells that were active during session A, sleep, and retrieval were not active in session F. Thus, these sub-ensembles indeed represent a context A-specific memory. The existence of these sub-ensembles results in the significant differences in population vector distance and matching score between session F and the other sessions, as shown in Fig. 4c, d and Fig. 5f, g. These new data have been added as Supplementary Figure 6, and we have added a statement in the revised manuscript, lines 15–18, page 12. The remaining sub-ensembles were active in session F, possibly representing commonalities in the experimental environment, such as the experimental room and experimenter, rather than just the context.

- Furthermore, the authors provide very little evidence that their task has induced learning (or a memory) at all. Without some sort of behavioral correlate of memory, it is unclear whether exposure to a context is sufficient to induce memory. And it is unclear whether active labelled cells represent anything at all.

In our original manuscript, we demonstrated that context exposure does indeed form contextual memories. In Fig. 1, context exposure under OFF-Dox induced Tet-Tag labelling of CA1 neurons, and the same context was later associated with shock. The shock memory can be retrieved by optical activation of the labelled cells (Fig. 1c, paired group), clearly indicating that context exposure formed an engram for contextual memory. The labelling protocol used for imaging was the same as that used for the experiment shown in Fig. 1. In addition, our group previously reported that simple context exposure can form a specific memory in hippocampus (Kitamura et al., 2012, *Mol Brain*, 5:5).

- I am concerned that supplementary Fig. 3B/C is not sufficiently stringent to provide convincing evidence of the authors' point. The authors claim that extracted patterns are composed of activity of constituent neurons. This comparison should be done using summary data from all patterns and their constituents, rather than simply providing one representative example for each case.

Although we would very much like to address the reviewer's comment, it is difficult to summarize the entire dataset in a figure because of the complexity of the cell components in each pattern and the large number of patterns.

Reviewer 3

General comments

In this paper, Ghandour and colleagues describe how they developed a novel imaging system and how this allowed them to study the activity patterns of engram and non-engram cells in vivo, during learning, subsequent sleep and recall. They show that, compared to non-engram cells, engram cells are more rhythmically activated during learning and that their activity is more stable across memory stages. Among engram cell populations, they identify sub-ensembles characterised by specific patterns of activity. The sub-ensembles that replay during sleep are also reactivated during retrieval.

This manuscript is interesting because it investigates the specific population dynamics of engram cells. The methods used to address this question are appropriate and sound. To the best of my knowledge this is the first reported use of the photoconvertible protein KikGR combined with calcium imaging to study memory engrams. Overall, the data presented here support the authors' conclusions.

The scientific narrative is well structured and flows naturally from one idea to the next. However, on multiple occasions, comprehension is hindered by grammar mistakes, poor word choices, and awkward sentence structures. Careful editing should allow this issue to be remedied.

In my opinion, the authors should contextualise their findings in the existing engram literature more thoroughly and emphasize what they believe makes this research novel.

We appreciate the reviewer's careful reading, positive feedback, and constructive comments, which have helped us improve the quality of the manuscript. We have sent the manuscript for English language editing to address the grammatical mistakes and other language flaws. We have also added a paragraph to the Discussion that emphasizes the main points of novelty in our paper (lines 6–13, page 15). We have taken into consideration all of the comments while revising the paper, as detailed below.

Figure 1

The experiments and results presented in this figure and the related text are convincing and in agreement with the established engram literature. The experimenters show that exposure to a context can induce freezing, when this context was previously paired with a shock. Optogenetically activating engram cells labelled in this context also induces freezing. I note that in Fig. 1 a), on the line for the unpaired condition, the word "freezing" is accompanied by an arrow pointing upwards, suggesting that freezing increases in this condition. However, as far as I understand, this is not the case (see Fig. 1 c)).

We thank the reviewer very much for noticing this mistake, which we have fixed in the revised manuscript.

Figure 2

The authors describe how they labelled and identified engram cells with KikGR and subsequently rendered this signal invisible for Ca²⁺ imaging. This is an elegant method that allows comparison between engram cell and non-engram cell activity in vivo.

We thank the reviewer for the positive feedback.

Figure 3

Thanks to a correlation matrix analysis, adequately described in the Methods section, this figure shows that engram cells exhibit more repetitive activity than non-engram cells during memory encoding. However, the authors make not attempt at explaining the rhythmic activity that also exists in non-engram cells. Could this be linked to hippocampal rhythms? I also would have liked to this this type of analysis for subsequent memory processing stages.

The reviewer raises a very interesting point. We agree that the rhythmic activity in the non-engram cells is interesting; however, it is not the main focus of the current work. Correlation matrix analysis detected the strong repetitive activity in the group of engram cells, indicating highly synchronous activity. In this paper, we were interested in studying the synchronous activity of engram cells in a deeper and more precise manner. That is why we shifted to non-negative matrix factorization, which detects synchronous ensembles and provides an indication of the degree of synchrony as well as which cells contribute to the synchronous activity during learning and in the subsequent memory processing stages (Fig. 5). This cannot be addressed by the correlation matrix analysis.

Figure 4

The result in this figure show that the activity of engram cells, as described by the Population Vector Difference, is more stable than that of non-engram cells. Fig. 4 d also highlights a significant difference between memory retrieval and exposure to a different context, but the text does not reflect on the relevance of that difference.

We have addressed the reviewer's point in the revised manuscript, lines 21–22, page 9, as follows:

“However this activity changed significantly upon moving from the retrieval to the different context, denoting the specificity of engram cell activity.”

Figure 5

Non-negative matrix factorization is an interesting method to identify and compare patterns of activity in cell populations. It allowed the authors to show that sub-ensembles of engram cells are characterised by specific patterns of activity that are repeated throughout experimental sessions, but less so when mice are exposed to a new context. In contrast, non-engram cells do not exhibit this tendency.

We thank the reviewer for the positive feedback.

Figure 6

Tracking activity patterns across sessions showed that engram cells that are reactivated during sleep are more often reactivated during retrieval compared to non-engram cells and that non-engram cells that are activated during learning are usually not reactivated during subsequent sessions.

On top of comparisons between engram and non-engram cells for each group, it would have been advisable to show statistical comparisons between groups (e.g. engram sub-ensembles reactivated during sleep and retrieval vs. during retrieval but not sleep).

As suggested, we have performed a further statistical analysis of the two groups within the engram population that display + + + and + - + patterns. However, there were no significant differences between the two groups for either NREM or REM sleep (paired *t*-test, two-tailed; NREM, $p = 0.34$; REM, $p = 0.74$). This is likely due to underdetection of sub-ensembles during sleep because we imaged only the first 1 min of each NREM and REM sleep cycle to avoid photobleaching, even though the total sleep time until the end of the first REM period is usually ~2 h. Thus, much of the sub-ensemble replay likely

occurred during the non-imaged periods of sleep; if we were able to image the entire replay, we might observe statistically significant differences between the + + + and + - + populations. We have mentioned this point in the revised manuscript (Discussion, lines 17–22, page 14; Fig. 6 legend).

In addition to the responses and changes described above, we have added precise information regarding the statistical analysis to each figure legend and to the Methods section because the previous version lacked this information.

Growth-Curve Analysis for Different Measures and Two Groups of Data

The growth curve is defined by

$$y_0(t) = \sum_n c_n t^n,$$

where $y(t)$ is the observation at time t and c_n is the coefficient of t^n . Growth curve analysis compares regressions on grouped (conditioned) data and ungrouped (unconditioned) data.

For regressions on unconditioned data, we fit a polynomial given by

$$y_i = c_0^{(0)} + c_1^{(0)} t_i + \dots + c_n^{(0)} t_i^n$$

on the data represented by (t_i, y_i) . We take $t_0 = 0$ to be session A, $t_0 = 1$ to be session B, and so on.

The order of the polynomial is determined by the Akaike information criterion with second-order correction (AICc):

$$AICc = 2k + N \ln \left(\frac{RSS}{N - k - 1} \right) + \frac{2k^2 + 2k}{N - k - 1},$$

where k is the number of coefficients, N is the number of data points, and RSS is the residual sum of squares.

For regressions on conditioned data, we fit a polynomial given by

$$y_i = c_0^{(0)} + c_1^{(0)} t_i + \dots + c_n^{(0)} t_i^n + q (c_0^{(1)} + c_1^{(1)} t_i + \dots + c_m^{(1)} t_i^m),$$

where q is the condition. Here, $q = 0$ for engram cells and $q = 1$ for non-engram cells. The order of the polynomial is also determined by the AICc. In this calculation, all regressions were calculated according to the least-squares method.

To show that the conditioned regression is superior, we conducted an F -test for regressions, as given by

$$F = \frac{\frac{RSS_{\text{unconditioned}} - RSS_{\text{conditioned}}}{m + 1}}{\frac{RSS_{\text{conditioned}}}{N - (n + m + 2)}}$$

The corresponding degrees of freedom are $(m + 1, N - (n + m + 2))$.

For the matching score between engram cells and non-engram cells, the fit performance for unconditional regression was:

n	RSS	AICc
0	8.76555	158.357
1	7.15161	145.822
2	4.85007	120.04
3	3.60811	100.986
4	3.58469	102.83
5	3.55163	104.545
6	3.55163	107.003

The results suggest that $n = 3$ provides the best fit for the unconditioned data. Then, conditioned regressions are performed for $m = 0, \dots, n$, with the following results:

m	RSS1	RSS2	AICc	d.f.	F-value	p-value
0	3.60811	2.12802	65.2829	(1,67)	46.6002	3.09×10^{-9}

1	3.60811	1.99682	63.0844	(2,66)	26.6286	3.32×10^{-9}
2	3.60811	1.87446	60.989	(3,65)	20.0391	2.62×10^{-9}
4	3.60811	1.86833	63.2891	(4,64)	14.8991	1.17×10^{-9}

The results suggest that $m = 2$ (i.e., quadratic perturbation) provides the best fit. Furthermore, the corresponding p -value is significant. This shows that the conditioned regression is significantly better than the unconditioned regression. We can thus conclude that cell type (the condition here) is an important factor.

Similarly, **for engram cells and shuffled engram cells**, the fit performance for unconditioned data is shown in the following table:

n	RSS	AICc
0	6.54255	137.297
1	5.41535	125.799
2	3.54617	97.4955
3	2.53593	75.5974
4	2.52606	77.6288
5	2.50133	79.3034
6	2.50133	81.7611

The fit results for conditioned data are:

m	RSS1	RSS2	AICc	d.f.	F-value	p-value
0	2.53593	2.19685	67.575	(1,67)	10.3413	0.0020
1	2.53593	2.17329	69.1818	(2,66)	5.50645	0.0061
2	2.53593	2.13333	70.3034	(3,65)	4.08891	0.0101
4	2.53593	2.13237	72.8065	(4,64)	3.02811	0.0240

The results suggest that $m = 0$ (i.e., vertical shifts) provides the best fit. Furthermore, the corresponding p -value is significant. This shows that the conditioned regression is significantly better than the unconditioned regression. We can thus conclude that shuffling (the condition here) is an important factor.

For the population vector distance between engram cells and non-engram cells, the fit performance of the unconditioned data is:

n	RSS	AICc
0	28.7766	243.946
1	17.542	210.425
2	17.2329	211.324
3	16.3425	209.749
4	14.9472	205.635
5	14.8062	207.336

6	14.8062	209.793
---	---------	---------

The results suggest that $n = 4$ provides the best fit for the unconditioned data. Then, conditioned regressions are performed for $m = 0, \dots, n$, with the following results:

m	RSS1	RSS2	AICc	d.f.	F-value	p-value
0	14.9472	12.4833	195.048	(1,67)	13.027	0.00059
1	14.9472	12.224	195.995	(2,66)	7.24034	0.0014
2	14.9472	12.0785	197.669	(3,65)	5.06671	0.0033
3	14.9472	12.0464	200.094	(4,64)	3.79271	0.0080
4	14.9472	11.939	202.153	(5,63)	3.12434	0.014

The results suggest that $m = 0$ (i.e., vertical shifts) provides the best fit. Furthermore, the corresponding p -value is significant. This shows that the conditioned regression is significantly better than the unconditioned regression. We can thus conclude that cell type (the condition here) is an important factor.

Reviewers' comments:

Reviewer #1 (Remarks to the Author):

The authors fully addressed my concerns.
Bong-Kiun kaang

Reviewer #2 (Remarks to the Author):

The authors have addressed several of my points, and the addition of figures and text has improved the manuscript. I very much think that the methods used by the authors are novel. However, I still have several serious concerns.

1) I remain confused about the nature of their data in Fig. 4B/C and Fig. 5F-I. In response to my previous review, the authors chose to perform a two-way repeated measures ANOVA. Based on their initial submission, though, I had presumed that their initial use of Wilcoxon sign-rank tests (a non-parametric equivalent to ANOVA) was because these data violated normality assumptions or other requirements for a parametric test.

A) Could the authors clarify whether their data, do in fact, satisfy the assumptions of normality required for this ANOVA? I agree that the ANOVA may be an appropriate test for their data, but only if these assumptions are satisfied.

B) Additionally, could the authors provide rationale for conducting a one-tailed analyses in Fig. 5 G & I? Of course, they should also correct for multiple comparisons in their post-hoc t-tests in these figures.

2) Regarding the shuffle controls in Fig. 5 H and I, the authors argue that higher activity is a prominent feature of engram cells and is therefore a relevant measurement. This may certainly be the case. However, the goal of the matching score analysis is to discover coordinated activity between neurons, and these types of correlation-based analyses can be easily confounded by different levels of total activity. If coordinated activity is of interest, then it is imperative that the authors control for the effects of total activity.

The authors showed that matching scores were higher than chance levels for engram neurons, however the comparison of interest is between the engram and non-engram populations. To make this comparison, the authors should first normalize engram and non-engram matching scores to their shuffled distributions. They could then make a plot similar to 5F, comparing the scores of normalized-engram to normalized-non-engram ensembles. Crucially, normalizing of both populations to a shuffled control would account for changes in total activity. This would allow the authors to examine coordinated activity differences between engram and non-engram ensembles. This is critical to support the authors' conclusions.

3) The authors' analysis of session F (Fig. 6 and S6) indicates that about half of their engram sub-ensembles are not context-selective. Instead, these sub-ensembles show activity even in a novel context. This poses a significant problem for the authors' analysis and conclusions. If these sub-ensembles do not encode the context, what is their role? Is it even appropriate to call them "engram" ensembles if they do not represent the tagged mnemonic event (context exposure)? One could just as easily argue that these ensembles are merely highly active across all sessions. But then, what precisely is labeled and why does it matter their activity during sleep? This problem is of deep concern goes to the central premise of the manuscript.

The authors seem to acknowledge this in the rebuttal letter, pointing out the differences in PVD

and matching score seem to be driven by the 50% of engram sub-ensembles that are context selective. However, the authors absolutely must address these concerns by parsing out context-selective vs. non-selective engram ensembles. What is an engram ensemble if not particular for the context? Without this analysis, readers can not be certain that their results are valid.

Reviewer #3 (Remarks to the Author):

- [Redacted]

Responses to reviewers' comments:

Reviewer #2 (Remarks to the Author):

The authors have addressed several of my points, and the addition of figures and text has improved the manuscript. I very much think that the methods used by the authors are novel. However, I still have several serious concerns.

1) I remain confused about the nature of their data in Fig. 4B/C and Fig. 5F-I. In response to my previous review, the authors chose to perform a two-way repeated measures ANOVA. Based on their initial submission, though, I had presumed that their initial use of Wilcoxon sign-rank tests (a non-parametric equivalent to ANOVA) was because these data violated normality assumptions or other requirements for a parametric test.

A) Could the authors clarify whether their data, do in fact, satisfy the assumptions of normality required for this ANOVA? I agree that the ANOVA may be an appropriate test for their data, but only if these assumptions are satisfied.

B) Additionally, could the authors provide rationale for conducting a one-tailed analyses in Fig. 5 G & I? Of course, they should also correct for multiple comparisons in their post-hoc t-tests in these figures.

We have checked the normality of the data and the data do satisfy the assumptions of normality required for the ANOVA. Please look at the file attached at the end of this reply (Normality test).

We previously performed a non-parametric test on the assumption that it's a stronger test, then we responded to the reviewer's request when he/she asked to do a test that accounts for several time points and applied the repeated measures ANOVA.

As we have mentioned in the manuscript in the statistics section, in some cases (Fig. 4 and Fig. 5) we chose to use one-tailed analysis. Because the engram cell activities were always higher than the non-engram activity, changes were expected only in one direction. However, two-tailed analyses also show the statistical significance and maintain the main conclusion driven from the data as shown below, but we continue to use one-tailed as it appears to be more appropriate in this case.

Fig. 4 p values (two-tailed):

(d) Difference in PVD between engram and non-engram cells. $n = 6$, paired t-test post-hoc tests, two-tailed. (Session B) $p = 0.00254$, (Session C) $p = 0.0058$, (Session D), $p = 0.0001526$, (Session E), $p = 0.01748$, (Session F), $p = 0.1038$.

- Significance is maintained in the two-tailed analysis

Fig.5 p values (two-tailed):

(g) Difference in MS between engram and non-engram cells across sessions. $n = 6$, paired t-test post-hoc tests, two-tailed, (Session B) $p = 0.00962$; (Session C) $p = 0.01162$; (Session D) $p = 0.01414$; (Session E) $p = 0.0041$; (Session F), $p = 0.0638$, (E vs F) $p = 0.0568$.

- Significance is maintained in the two-tailed analysis, except (Session F) and (E vs F). However, this does not change the conclusion, as in session (F) there is no statistically significant difference between engram and non-engram cells. This infers that engram cell activities change in the different context as the non-engram cells.

(i) Difference in MS between original and shuffled engram data across sessions. $n = 6$, paired t-test post-hoc tests, two-tailed, (Session B) $p = 0.0414$; (Session C) $p = 0.392$; (Session D) $p = 0.068$; (Session E) $p = 0.00368$; (Session F) $p = 0.338$.

- Significance is maintained in the two-tailed analysis, except (A vs D) However, tendency is clear, and the main conclusion is maintained.

2) Regarding the shuffle controls in Fig. 5 H and I, the authors argue that higher activity is a prominent feature of engram cells and is therefore a relevant measurement. This may certainly be the case. However, the goal of the matching score analysis is to discover coordinated activity between neurons, and these types of correlation-based analyses can be easily confounded by different levels of total activity. If coordinated activity is of interest, then it is imperative that the authors control for the effects of total activity.

The authors showed that matching scores were higher than chance levels for engram neurons, however the comparison of interest is between the engram and non-engram populations. To make this comparison, the authors should first normalize engram and non-engram matching scores to their shuffled distributions. They could then make a plot similar to 5F, comparing the scores of normalized-engram to normalized-non-engram ensembles.

Crucially, normalizing of both populations to a shuffled control would account for changes in total activity. This would allow the authors to examine coordinated activity differences between engram and non-engram ensembles. This is critical to support the authors' conclusions.

According to the reviewer's request, we have normalized the activity of both engram and non-engram cells and compared their normalized activity. Normalized engram data shows significantly higher MS across sessions compared with normalized non-engram data. This result is added as new Supplementary Fig. 6 with statistical information as below. Collectively, all data of the MS analysis support our conclusion that, compared with non-engram, engram shows higher possibility in sub-ensembles reappearance across sessions.

Supplementary Fig. 6:

MS across sessions (with respect to session A) for normalized engram and normalized non-engram data. $n = 6$, two-way repeated-measures ANOVA, $p = 0.0273$ (between cell types). Post-hoc paired t-test tests; (E vs F, in engram) $p = 0.0235$.

3) The authors' analysis of session F (Fig. 6 and S6) indicates that about half of their engram sub-ensembles are not context-selective. Instead, these sub-ensembles show activity even in a novel context. This poses a significant problem for the authors' analysis and conclusions. If these sub-ensembles do not encode the context, what is their role? Is it even appropriate to call them "engram" ensembles if they do not represent the tagged mnemonic event (context exposure)? One could just as easily argue that these ensembles are merely highly active across all sessions. But then, what precisely is labeled and why does it matter their activity during sleep? This problem of deep concern goes to the central premise of the manuscript.

The authors seem to acknowledge this in the rebuttal letter, pointing out the differences in PVD and matching score seem to be driven by the 50% of engram sub-ensembles that are context selective. However, the authors absolutely must address these concerns by parsing out context-selective vs. non-selective engram ensembles. What is an engram ensemble if not particular for the context? Without this analysis, readers cannot be certain that their results are valid.

The reviewer wondered why the several sub-ensembles in the engram population showed their activity even in the different context while others were active only in the original context. The specific engram sub-ensembles, which were active only for the retrieval in the original context, were indeed responsible for the specific representation of the square context only. Other shared engram sub-ensembles, which were active in both contexts, also should have correspondence to the session A because all engram ensembles were labelled at the first experience of session A. The shared engram sub-ensembles may represent the commonalities between 2 contexts, i.e. bigger environmental context such as experimental room and experimenter.

A previous paper using Arc CatFISH, by Guzowski et al 1999, Nat. Neuroscience, showed that a significant amount of CA1 neurons were double positive for context A and B with a percentage around 40% of the totally activated neurons in context A¹. In addition, a recently published study, Tanaka et al Science 2018, indicated that when animals explored a different context 43.5% of the engram cells had peak firing rate less than 1 Hz², indicating that the remaining half of the engram cells, 56.5%, were active in both contexts. Collectively, these evidences support the notion that genetically labelled CA1 engram cells may represent information corresponding to a specific context and an environmental commonality by using different sets of sub-ensembles.

This point is discussed in the Discussion, page 15.

Reviewer #3 (Remarks to the Author):

[Redacted]

References:

- 1 Guzowski, J. F., McNaughton, B. L., Barnes, C. A. & Worley, P. F. Environment-specific expression of the immediate-early gene Arc in hippocampal neuronal ensembles. *Nature Neuroscience* **2**, 1120, doi:10.1038/16046 (1999).
- 2 Tanaka, K. Z. *et al.* The hippocampal engram maps experience but not place. *Science* **361**, 392 (2018).

Normality Tests

Population Distance

In order to check if the assumption for t-test and ANOVA is valid, a series of Kolmogorov-Smirnov Tests of Normality was performed on residuals of different data groups.

Data Set	p-value
Engram Cells, Session B	0.90816
Engram Cells, Session C	0.4983
Engram Cells, Session D	0.74738
Engram Cells, Session E	0.20285
Engram Cells, Session F	0.46368
Non-engram Cells, Session B	0.91243
Non-engram Cells, Session C	0.93868
Non-engram Cells, Session D	0.95483
Non-engram Cells, Session E	0.71284
Non-engram Cells, Session F	0.85702
Residuals of all data in ANOVA	0.08656

Form the series of normality tests, there is no data set having a significant difference with the closest normal distribution. The result suggests that the normality assumption for the tests is valid.

Matching Scores

The normality tests are also performed for normalized matching for different test and ANOVA to see if the assumption is valid.

Data Set	p-value
Engram Cells, Normalized, Session B	0.81445

Engram Cells, Normalized, Session C	0.95395
Engram Cells, Normalized, Session D	0.49286
Engram Cells, Normalized, Session E	0.83599
Engram Cells, Normalized, Session F	0.91314
Non-engram Cells, Normalized, Session B	0.29559
Non-engram Cells, Normalized, Session C	0.54128
Non-engram Cells, Normalized, Session D	0.48705
Non-engram Cells, Normalized, Session E	0.67819
Non-engram Cells, Normalized, Session F	0.94914
Residuals of all data in ANOVA	0.39727

Here we can see that the normality assumption for various tests is valid.

The normality tests have also been done on matching scores of original data and shuffled data.

Data Set	p-value
Original Engram Cells, Session B	0.8454
Original Engram Cells, Session C	0.83291
Original Engram Cells, Session D	0.96428
Original Engram Cells, Session E	0.83599
Original Engram Cells, Session F	0.83108
Shuffled Engram Cells, Session B	0.97699
Shuffled Engram Cells, Session C	0.98464
Shuffled Engram Cells, Session D	0.95568
Shuffled Engram Cells, Session E	0.95093
Shuffled Engram Cells, Session F	0.99744

Original Non-engram Cells, Session B	0.76177
Original Non-engram Cells, Session C	0.53037
Original Non-engram Cells, Session D	0.71666
Original Non-engram Cells, Session E	0.95683
Original Non-engram Cells, Session F	0.86857
Shuffled Non-engram Cells, Session B	0.29224
Shuffled Non-engram Cells, Session C	0.56306
Shuffled Non-engram Cells, Session D	0.63202
Shuffled Non-engram Cells, Session E	0.62769
Shuffled Non-engram Cells, Session F	0.88373

Here we can see that the normality assumption for various tests is valid.

Reviewers' comments:

Reviewer #2 (Remarks to the Author):

I have re-read this manuscript and it is improved from the original version. A few lingering points remain.

1) Although the authors have provided sufficient evidence that their data are normally distributed (and that an ANOVA therefore, is the most appropriate mechanism to analyze these data), they still have not addressed my concerns from the last revision regarding multiple post-hoc comparisons. The standard in the field is to correct for multiple post-hoc comparisons. Therefore, the post-hoc tests in Fig 4D, 5G and 5I are not appropriate. Elsewhere in the manuscript (Fig 1B, 1C) the authors use Scheffe's method to conduct post-hoc tests (this is the appropriate post-hoc test for multiple comparisons). At the very least, the authors should do the same for their other post-hoc tests.

2) I am satisfied with the authors' new Figure S6 which adequately tests differences between engram and non-engram populations. As with my above point, the authors should use appropriate post-hoc tests to analyze their data, and not simply use multiple uncorrected t-tests.

3) I remain unconvinced by the authors' conclusions about their context non-selective tagged ensembles. They argue that these purported engram neurons are encoding broader contextual similarities (experimental room, human experimenters, etc.), which explains their activity in both experimental contexts. If this is indeed the authors' position, they must be clearer about how their data fit with this view, up front. The added section in the discussion (page 15) should make specific reference to relevant findings from this study (i.e. ~50% of engram neurons are non-specifically active in both contexts).

Reviewers' comments:

Reviewer #2:

I have re-read this manuscript and it is improved from the original version. A few lingering points remain.

1) Although the authors have provided sufficient evidence that their data are normally distributed (and that an ANOVA therefore, is the most appropriate mechanism to analyze these data), they still have not addressed my concerns from the last revision regarding multiple post-hoc comparisons. The standard in the field is to correct for multiple post-hoc comparisons. Therefore, the post-hoc tests in Fig 4D, 5G and 5I are not appropriate. Elsewhere in the manuscript (Fig 1B, 1C) the authors use Scheffe's method to conduct post-hoc tests (this is the appropriate post-hoc test for multiple comparisons). At the very least, the authors should do the same for their other post-hoc tests.

According to the reviewer's suggestion, we have conducted Bonferroni's test as post-hoc tests in Fig. 4d, 5g, and 5i, because Bonferroni's test is normally used and appropriate for the multiple comparison. The results were very similar to the previous results of paired t-test except for Fig 4d.

In Fig 4d, there is a significant difference between engram and non-engram in session F. Therefore, we erased one sentence from the interpretation of the PVD in Result section, "However, this activity changed significantly upon moving from the retrieval context to a different context, reflecting the specificity of engram-cell activity (Fig. 4d).". This does not affect the main conclusion of this manuscript that "a single engram population is composed of several sub-ensembles", because this conclusion is fully supported by the results of NMF analyses in Fig. 5g and 5i.

We have revised the statistical analysis part in Methods section and figure legends.

2) I am satisfied with the authors' new Figure S6 which adequately tests differences between engram and non-engram populations. As with my above point, the authors should use appropriate post-hoc tests to analyze their data, and not simply use multiple

uncorrected t-tests.

We have performed Bonferroni's post-hoc test as described in responses to the comment 1. We do not see significant difference between engram and non-engram in each session. However, this supplemental data focuses on the difference between engram and non-engram cells, not interaction of cell types and sessions. Two-way RM ANOVA had already showed the significant difference between two cell types. Therefore, we keep the last conclusion, “Even after normalizing both engram and non-engram data by their shuffled activity, engram cells maintained their higher MS across sessions (Supplementary Figure 6)”, in this revision.

3) I remain unconvinced by the authors' conclusions about their context non-selective tagged ensembles. They argue that these purported engram neurons are encoding broader contextual similarities (experimental room, human experimenters, etc.), which explains their activity in both experimental contexts. If this is indeed the authors' position, they must be clearer about how their data fit with this view, up front. The added section in the discussion (page 15) should make specific reference to relevant findings from this study (i.e. ~50% of engram neurons are non-specifically active in both contexts).

We do not conclude and demonstrate in the manuscript that “context non-selective tagged ensembles encode broader contextual similarities.” We just suggest this possibility to interpret the results in the Discussion section.